# LEARNING TO RANK CHAIN-OF-THOUGHT: USING A SMALL MODEL

## ABSTRACT

Large Language Models (LLMs) struggle with reliable mathematical reasoning, and current verification methods are often computationally expensive. This paper introduces the Energy Outcome Reward Model (EORM), a highly efficient, lightweight post-hoc verifier designed to address this challenge. EORM uses an energy-based framework to rank Chain-of-Thought (CoT) solutions, learning to distinguish correct from incorrect reasoning using only simple outcome labels, thus eliminating the need for expensive annotations. With only 55M parameters, over 127 times smaller than typical reward models, EORM boosts the accuracy of Llama 3 8B to 90.7% on GSM8k and 63.7% on MATH. This performance is achieved by efficiently selecting the optimal reasoning path from a pool of candidates, allowing it to match or exceed the accuracy of far more resource-intensive Best-of-N sampling techniques. Crucially, our experiments show that EORM generalizes effectively to out-of-distribution problems and unseen models, indicating it learns fundamental principles of valid reasoning. This robustness, combined with its efficiency, establishes EORM as a practical tool for deploying more dependable LLMs in complex, real-world applications. Our code is available: `https://anonymous.4open.science/r/Learning-to-rank-COT-anynomized-7718/`

## 1 INTRODUCTION

Mathematical reasoning remains a critical and challenging domain for Large Language Models (LLMs), demanding a high degree of logical consistency and multi-step inferential accuracy that often eludes current architectures (Guo et al., 2025; Tong et al., 2024). While Chain-of-Thought (CoT) (Wei et al., 2022; Kojima et al., 2022) has significantly advanced the ability of LLMs to articulate intermediate reasoning steps, thereby improving performance on complex tasks (Wang et al., 2023; Huang et al., 2022; Yao et al., 2022), it offers no intrinsic guarantee of correctness. This leads to a fundamental challenge: among the multiple Chain-of-Thought produced by the LLMs, how can we reliably identify the single most accurate one? This selection problem, widely referred to as the *Best-of-N challenge*, has become a key bottleneck in advancing the reliability of mathematical reasoning with LLMs (Tong et al., 2024; Xiong et al., 2024; Press et al., 2022).

To solve the Best-of-N problem, researchers have tried many approaches, and currently, existing approaches fall into two categories: (1) majority voting and (2) post-training with reinforcement learning (RL) to train a reward model for answer selection. Majority voting (Toh et al., 2025; Wang et al., 2023) is the simplest method, it chooses the answer most frequently generated by the model. While this method is easy to implement, it tends to reflect the model's biases rather than identifying the truly correct answer. The RL-based approaches are more sophisticated, with three prominent variants: **(1) Preference Optimization (PO)**, **(2) Outcome Reward Models (ORM)**, and **(3) Process Reward Models (PRM)**. Preference Optimization (Pang et al., 2024; Zhang et al., 2024) uses pairwise comparisons between correct and incorrect answers to train a reward model. Although it can improve answer quality, it requires training a new model for each task and struggles with scenarios involving multiple complex outputs. Outcome Reward Models (Lyu et al., 2025; Hosseini et al., 2024; Cobbe et al., 2021b) are similar but allow supervision from multiple positive and negative examples, making them more flexible than pairwise-only methods. Process Reward Models (She et al., 2025; Wang et al., 2025) go one step further by assigning rewards to each step in the reasoning process. However, this requires extensive annotation of every Chain-of-Thought

| Feature | ORM | PRM | PO | EORM (ours) |
|---|---|---|---|---|
| Does not require Step-by-Step Labels? | ✓ | ✗ | ✓ | ✓ |
| Does not require Preference Pair Labels? | ✓ | ✓ | ✗ | ✓ |
| Uses Binary Outcome Labels as Reward? | ✗ | ✗ | ✗ | ✓ |
| Low Annotation Cost? | ✓ | ✗ | ✓ | ✓ |
| Lightweight Model | ✗ | ✗ | ✗ | ✓ |

Table 1: **EORM's feature benefit compared to other reasoning baselines.** We compare EORM with ORM (Outcome Reward Model), PRM (Process Reward Model), and PO (Preference Optimization) to demonstrate the feature benefit of our method over these baselines.

(CoT) trace, making data preparation labor-intensive. We provide a more in-depth comparison of the limitations of the different types of reward model in Table 1.

This motivates the need for a more efficient and generalizable alternative. Our approach begins with the observation that model outputs are sequences of tokens, which allows us to tokenize these responses and analyze their statistical patterns. Each output can be interpreted as a sample from an underlying probability distribution. Leveraging the Universal Approximation Theorem (Liu et al., 2025; Kratsios, 2021), we recognize that a sufficiently expressive neural network can, in principle, learn to distinguish between these distributions. Building on this insight, we propose training a lightweight neural network model to classify responses as good or bad, effectively select the best answer using Best-of-N, acting like a "LLM-as-a-Judge". Crucially, because this model learns from token-level patterns rather than task-specific features, it is capable of generalizing across outputs from different base LLMs. To further enhance our neural network ability to capture complex distributions and support robust generalization, we adopt an Energy-Based Model (EBM) architecture (Deng et al., 2020b; Du & Mordatch, 2020). EBMs have demonstrated strong performance in related tasks, utilizing their uniqueness of using Energy Probabilistic Landscape, and in our experiments, our EBM-based judge consistently outperforms both majority voting and traditional Outcome Reward Models in identifying the best answer.

In this paper, we introduce the Energy Outcome Reward Model (EORM), an efficient post-hoc verifier for CoT outputs. EORM leverages principles from Energy-Based Models (EBMs) (Du & Mordatch, 2019; Liu et al., 2020) to effectively rerank and select the best candidate solutions. Our methodology trains a lightweight model to assign a scalar energy score to each candidate, learning to distinguish correct from incorrect reasoning using only simple binary outcome labels. Critically, this approach allows us to drastically reduce the model size from the typical 7B to 8B parameters of reward models to just 55M, achieving a greater than 127x reduction in parameter count while maintaining high performance.

Our main contributions are threefold:

- **A Novel Small and Efficient Energy Reward Model for CoT Verification.** We propose EORM, a lightweight and efficient verifier that assigns a scalar energy score to each CoT solution. By training on simple binary outcome labels (correct/incorrect), EORM eliminates the need for costly step-by-step annotations or preference pairs required by Process and Preference-based Reward Models.

- **State-of-the-Art Performance on Math Reasoning.** We demonstrate that EORM significantly boosts the performance of various open-source LLMs on challenging mathematical reasoning benchmarks, including GSM8k and MATH. Our approach consistently achieves superior accuracy by effectively reranking candidate solutions, often matching or exceeding the performance of more computationally intensive methods.

- **Demonstrated Robust Generalization.** We empirically validate that EORM generalizes effectively to both out-of-distribution datasets and unseen LLM architectures. This highlights its ability to learn underlying principles of logical reasoning rather than overfitting to the stylistic patterns of the models in its training set.

The remainder of this paper is organized as follows. We first review related work in Section 2. Next, we detail the EORM architecture and training methodology in Section 3. We then present our comprehensive experimental results and analysis in Section 4 and abaltions in Section 5. Finally, we conclude the paper in Section 6.

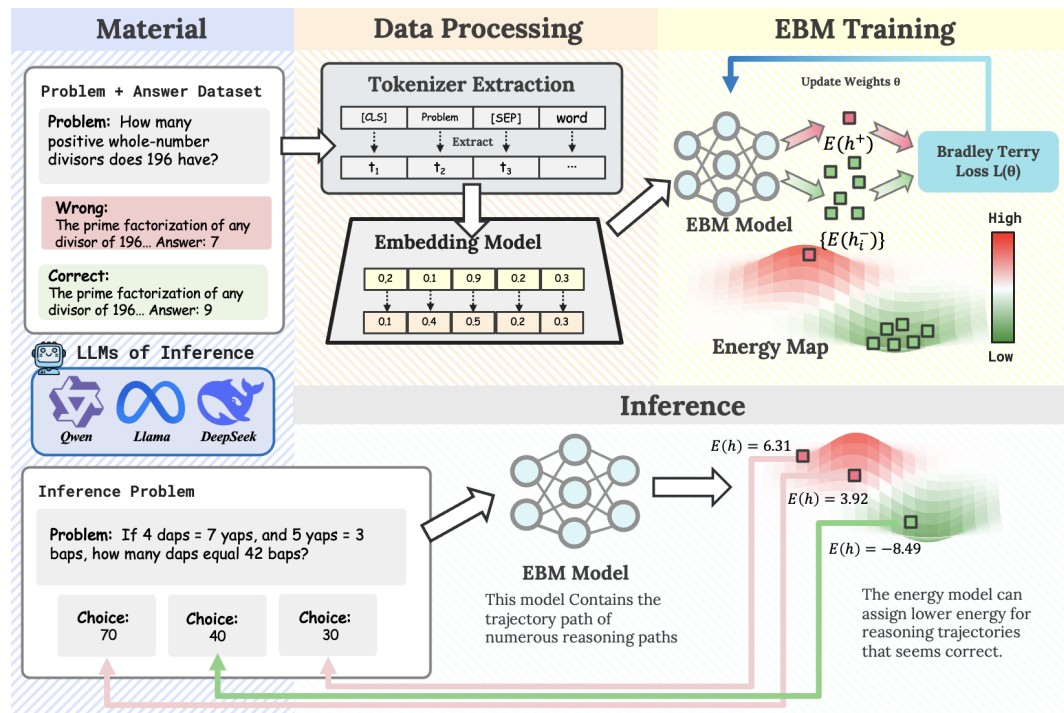

Figure 1: **An overview of flow chart of EORM.** In the EORM process, the model tokenizes the question-answer pair, then computes an energy score using an Energy-Based Model (EBM). The Bradley-Terry loss serves as the objective for reward-based fine-tuning. During deployment, the trained energy reward model computes energy scores for classification tasks.

## 2 RELATED WORK

Our work intersects with advancements in Chain-of-Thought (CoT) reasoning, the verification and reranking of Large Language Model (LLM) outputs, and Energy-Based Models (EBMs). CoT reasoning (Wei et al., 2022; Kojima et al., 2022) has significantly improved multi-step reasoning in LLMs (Bai et al., 2023; Touvron et al., 2023; OpenAI, 2023; Guo et al., 2025), with further refinements like Least-to-Most prompting (Zhou et al., 2022) and Tree-of-Thoughts (Yao et al., 2023; Zhang et al., 2023). However, the fallibility of CoT outputs (Tong et al., 2024), where a single error can invalidate a solution, necessitates robust verification. Common approaches like self-consistency (Wang et al., 2023) improve reliability but incur substantial computational costs by sampling numerous solutions (Wu et al., 2024). To address this, various reranking and verification techniques have emerged, including training separate verifier models (Cobbe et al., 2023; Khalifa et al., 2023; Li et al., 2023) or adopting learning-to-rank perspectives (Liu, 2009; Deng et al., 2020a). These methods, while effective, often introduce their own complexities or computational demands. Our approach, EORM, draws inspiration from EBMs (Du & Mordatch, 2019), which assign scalar energy scores to configurations and are well-suited for ranking (Grathwohl et al., 2019; Liu, 2009). Specifically, we leverage the insight that classifier logits can be interpreted as negative energies (Grathwohl et al., 2019; Liu et al., 2020). By training a lightweight EBM with a pairwise Bradley-Terry loss (Liu, 2009) on outcome labels, EORM efficiently reranks CoT candidates, offering a distinct, streamlined post-hoc verification mechanism that complements retrieval-augmented methods (Rakin et al., 2024; Schick et al., 2023; Yao et al., 2022; Press et al., 2022; Khattab et al., 2022) and other specialized verifiers like Chain-of-Actions (Pan et al., 2025) or Search-in-the-Chain (Xu et al., 2023). Recent work by Piotrowski et al. (2025); Li et al. (2025) also explores efficient verifiers but relies on extracting hidden states from the base LLM. In contrast, EORM operates directly on the token outputs, offering greater flexibility without requiring access to the generator's internal states. For a more detailed discussion of related literature, please refer to Appendix B.

# 3   METHODOLOGY

In this section, we introduce the architectural design and technical foundations of our proposed **Energy Outcome Reward Model (EORM)**, a lightweight verifier for Chain-of-Thought (CoT) reasoning. We begin by outlining the principles of Energy-Based Models (EBMs), which form the foundation of our approach. We then detail the specific architecture and training objective of EORM, showing how it learns to rank CoT candidates effectively using only simple outcome-based supervision.

## 3.1   PRELIMINARIES: ENERGY-BASED MODELS FOR RANKING

Evaluating CoT reasoning requires a flexible mechanism that can distinguish high-quality reasoning from incorrect or incoherent paths. Instead of treating this as a classification task, we model it as a preference-ranking problem using Energy-Based Models (EBMs).

Energy-Based Models (EBMs) provide a flexible approach to modeling distributions by assigning a scalar energy $E_\theta(y)$ to each configuration $y$ from a space $\mathcal{Y}$. This energy function is parameterized by $\theta$. Lower energy typically corresponds to more desirable or probable configurations. In this work, $\mathcal{Y}$ is the space of possible Chain-of-Thought text sequences $y$, and $\theta$ represents the learnable parameters of our EORM model.

Given an energy function $E_\theta(y)$, the corresponding Boltzmann (or Gibbs) distribution $p_\theta(y)$ assigns a probability to each configuration $y \in \mathcal{Y}$ as:

$$p_\theta(y) \;=\; \frac{\exp\big(-E_\theta(y)\big)}{Z_\theta}, \tag{1}$$

where $Z_\theta$ is the partition function, a normalization constant that ensures the probability distribution sums to unity:

$$Z_\theta \;=\; \sum_{y' \in \mathcal{Y}} \exp\big(-E_\theta(y')\big) \quad \text{(for discrete } \mathcal{Y}\text{)}. \tag{2}$$

A key challenge in working with EBMs is that computing the partition function $Z_\theta$ is often computationally intractable. However, for tasks involving ranking or selecting the best candidate from a finite set $\mathcal{Y}_{\text{cand}} \subset \mathcal{Y}$, the explicit computation of $Z_\theta$ is unnecessary. Since $Z_\theta$ is constant for all $y \in \mathcal{Y}_{\text{cand}}$, comparing probabilities $p_\theta(y)$ is equivalent to comparing the unnormalized scores $\exp(-E_\theta(y))$, which in turn relates directly to comparing the energies $E_\theta(y)$. This equivalence implies that energy minimization provides a direct mechanism for identifying the most probable (preferred) solution within a candidate set, without needing to compute $Z_\theta$:

$$y^* \;=\; \arg\min_{y \in \mathcal{Y}_{\text{cand}}} E_\theta(y) \;=\; \arg\max_{y \in \mathcal{Y}_{\text{cand}}} p_\theta(y). \tag{3}$$

This makes EBMs well-suited for our task of reranking CoT solutions based on learned preferences encoded in the energy function $E_\theta(y)$.

## 3.2   EORM: ARCHITECTURE AND TRAINING OBJECTIVE

Building on the EBM framework, EORM is designed as a lightweight yet powerful verifier for CoT reasoning. The core idea is to train a small neural network to learn an energy function $E_\theta(y)$ that maps any given CoT solution $y$ to a scalar energy score. By design, this function assigns low energy to correct reasoning paths and high energy to incorrect ones, enabling effective reranking of candidate solutions.

**Architecture.** To implement the energy function $E_\theta(y)$, EORM uses a lightweight Transformer encoder. A given CoT solution $y$ is first tokenized, and a special classification token (e.g., `[CLS]`) is prepended. The sequence is then passed through the Transformer encoder. The final hidden state corresponding to the `[CLS]` token, denoted $\mathbf{h}_{\text{CLS}}$, serves as a holistic representation of the entire reasoning path. This representation is fed into a simple energy head—a Multi-Layer Perceptron (MLP) with Layer Normalization—which projects it to the final scalar energy value:

$$E_\theta(y) \;=\; \text{MLP}\Big(\text{LayerNorm}\big(\mathbf{h}_{\text{CLS}}\big)\Big) \in \mathbb{R}. \tag{4}$$

This scalar output $E_\theta(y)$ represents the energy assigned to the sequence $y$. Lower values indicate higher assessed quality or correctness.

**Training Objective.** EORM is trained to distinguish between correct and incorrect solutions using a simple yet effective pairwise ranking loss. This approach only requires binary outcome labels for each CoT, avoiding the need for expensive, fine-grained annotations required by process-based reward models. For a given problem, we collect a set of generated CoT solutions $\mathcal{Y}_n$ and partition it into a subset of correct solutions, $\mathcal{Y}_+$, and a subset of incorrect solutions, $\mathcal{Y}_-$.

$$\mathcal{Y}_+ \;=\; \big\{ y \in \mathcal{Y}_n \mid l(y) = 1\big\}, \quad \mathcal{Y}_- \;=\; \big\{ y \in \mathcal{Y}_n \mid l(y) = 0\big\}. \tag{5}$$

The model's parameters $\theta$ are then optimized to ensure that for any pair of solutions $(y_+, y_-)$ where $y_+ \in \mathcal{Y}_+$ and $y_- \in \mathcal{Y}_-$, the energy of the correct solution is lower than that of the incorrect one. We formalize this using the Bradley-Terry loss, which is summed over all possible pairs within the group:

$$\mathcal{L}(\theta; \mathcal{Y}_n) \;=\; \frac{1}{|\mathcal{Y}_+||\mathcal{Y}_-|} \sum_{y_+ \in \mathcal{Y}_+} \sum_{y_- \in \mathcal{Y}_-} \log\Big(1 + \exp\big(E_\theta(y_+) - E_\theta(y_-)\big)\Big). \tag{6}$$

Minimizing this loss encourages a clear separation in the energy landscape, pushing the model to assign consistently lower energy scores to correct solutions. This pairwise formulation provides a strong learning signal, particularly beneficial for CoT ranking where subtle differences can exist between multiple flawed or correct reasoning paths. During training, the model parameters $\theta$ are updated by iterating through the dataset and minimizing this loss via gradient-based optimization. The complete training procedure is summarized in Algorithm 1 For a more detailed theoretical analysis, please refer to Appendix C

**Implementation.** It is crucial to distinguish between the training and deployment phases of EORM. During the **offline training phase**, the model requires ground-truth labels to learn the energy landscape that distinguishes valid reasoning. However, during the **online deployment phase**, EORM functions as a standalone verifier for open-domain problems where ground-truth answers are unavailable. By internalizing the fundamental principles of validity during training, EORM can rank and select the most plausible solution from unlabeled candidates at inference time.

## 4 EXPERIMENTS

In this section, we empirically evaluate our proposed Energy Outcome Reward Model (EORM). A key advantage of EORM, as highlighted in our introduction, is its exceptional efficiency; with only 55M parameters, it is over 127 times smaller than typical 7B reward models, as shown in Figure 2. Our evaluation is structured into three main parts. **(1)** First, in Section 4.2, we assess its performance on in-distribution mathematical reasoning tasks. **(2)** Second, in Section 4.3, we test its robustness on out-of-distribution benchmarks. **(3)** Finally, in Section 4.4, we analyze its ability to generalize to outputs from unseen LLM architectures.

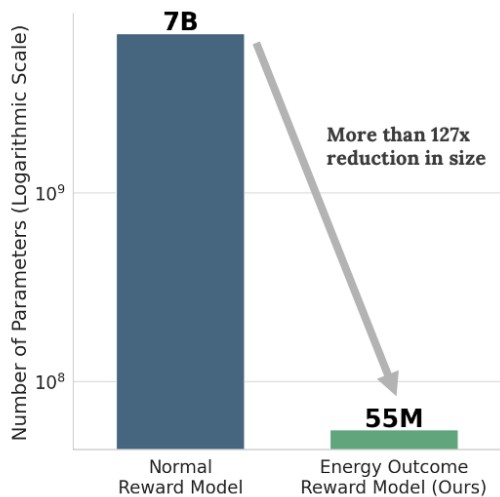

Figure 2: **A comparison of the parameter sizes between a standard reward model and our EORM Model.** A typical reward model has approximately 7 billion parameters, while EORM has only 55 million, demonstrating a size reduction of over 127 times and highlighting EORM's efficiency.

### 4.1 EXPERIMENTAL SETUP

For our experiments, we use the GSM8k (Cobbe et al., 2021a) and MATH (Hendrycks et al., 2021) datasets for in-distribution tasks, and AIME 2024, AMC, and AGIEval's SAT Math and Gaokao Math subsets (Zhong et al., 2023) for out-of-distribution evaluation. We applied EORM to several open-source Large Language Models (LLMs): Mistral-7B-v0.1 (Jiang et al.,

| Model | Base Model | Params | GSM8k | MATH | Avg. |
|---|---|---|---|---|---|
| Mistral-v0.1 (Jiang et al., 2023) | Mistral-v0.1 | 7B | $42.9_{\blacktriangle+0.0}$ | $12.9_{\blacktriangle+0.0}$ | 27.9 |
| MathScale (Tang et al., 2024) | Mistral-v0.1 | 7B | $74.8_{\blacktriangle+31.9}$ | $35.2_{\blacktriangle+22.3}$ | 55.0 |
| MMIQC (Liu et al., 2023) | Mistral-v0.1 | 7B | $74.8_{\blacktriangle+31.9}$ | $36.0_{\blacktriangle+23.1}$ | 55.4 |
| MetaMath (Yu et al., 2024) | Mistral-v0.1 | 7B | $77.9_{\blacktriangle+35.0}$ | $28.6_{\blacktriangle+15.7}$ | 53.3 |
| KPMath-Plus (Huang et al., 2024) | Mistral-v0.1 | 7B | $82.1_{\blacktriangle+39.2}$ | $46.8_{\blacktriangle+33.9}$ | 64.5 |
| DART-Math (Tong et al., 2024) | Mistral-v0.1 | 7B | $82.6_{\blacktriangle+39.7}$ | $43.5_{\blacktriangle+30.6}$ | 63.1 |
| Math-Shepherd (Wang et al., 2024) | Mistral-v0.1 | 7B | $84.1_{\blacktriangle+41.2}$ | $33.0_{\blacktriangle+20.1}$ | 58.6 |
| MAmmoTH2-Plus (Yue et al., 2024) | Mistral-v0.1 | 7B | $84.7_{\blacktriangle+41.8}$ | $45.0_{\blacktriangle+32.1}$ | 64.9 |
| Xwin-Math (Li et al., 2024a) | Mistral-v0.1 | 7B | $89.2_{\blacktriangle+46.3}$ | $43.7_{\blacktriangle+30.8}$ | 66.5 |
| WizardMath-Mistral (Luo et al., 2025) | Mistral-v0.1 | 7B | $90.7_{\blacktriangle+47.8}$ | $55.4_{\blacktriangle+42.5}$ | 73.1 |
| **EORM (Ours)** | **Mistral-v0.1** | 7B | $\mathbf{91.0}_{\blacktriangle+48.1}$ | $\mathbf{48.8}_{\blacktriangle+35.9}$ | **69.9** |
| Llama2 (Touvron et al., 2023) | Llama 2 | 7B | $14.6_{\blacktriangle+0.0}$ | $2.5_{\blacktriangle+0.0}$ | 8.6 |
| MAmmoTH-CoT (Yue et al., 2024) | Llama 2 | 7B | $50.5_{\blacktriangle+35.9}$ | $10.4_{\blacktriangle+7.9}$ | 30.5 |
| MetaMath (Yu et al., 2024) | Llama 2 | 7B | $66.5_{\blacktriangle+51.9}$ | $19.8_{\blacktriangle+17.3}$ | 43.2 |
| Math-Shepherd (Wang et al., 2024) | Llama 2 | 7B | $73.2_{\blacktriangle+58.6}$ | $21.6_{\blacktriangle+19.1}$ | 47.4 |
| **EORM (Ours)** | **Llama 2** | 7B | $\mathbf{75.6}_{\blacktriangle+61.0}$ | $\mathbf{21.8}_{\blacktriangle+19.3}$ | **48.7** |
| DeepSeekMath-Base (Shao et al., 2024) | DeepSeekMath | 7B | $64.2_{\blacktriangle+0.0}$ | $36.2_{\blacktriangle+0.0}$ | 50.2 |
| NuminaMath-CoT (Li et al., 2024b) | DeepseekMath | 7B | $75.4_{\blacktriangle+11.2}$ | $55.2_{\blacktriangle+19.0}$ | 65.3 |
| MMIQC (Liu et al., 2023) | DeepSeekMath | 7B | $79.0_{\blacktriangle+14.8}$ | $45.3_{\blacktriangle+9.1}$ | 62.2 |
| KPMath-Plus (Huang et al., 2024) | DeepSeekMath | 7B | $83.9_{\blacktriangle+19.7}$ | $48.8_{\blacktriangle+12.6}$ | 66.4 |
| DeepSeekMath-RL (Shao et al., 2024) | DeepSeekMath | 7B | $88.2_{\blacktriangle+24.0}$ | $51.7_{\blacktriangle+15.5}$ | 70.0 |
| **EORM (Ours)** | **DeepSeekMath** | 7B | $\mathbf{84.2}_{\blacktriangle+20.0}$ | $\mathbf{58.7}_{\blacktriangle+22.5}$ | **71.5** |
| Llama 3 (Team, 2024) | Llama 3 | 8B | $76.6_{\blacktriangle+0.0}$ | $28.9_{\blacktriangle+0.0}$ | 52.8 |
| MetaMath (Yu et al., 2024) | Llama 3 | 8B | $77.3_{\blacktriangle+0.7}$ | $30.8_{\blacktriangle+1.9}$ | 54.1 |
| MMIQC (Liu et al., 2023) | Llama 3 | 8B | $77.6_{\blacktriangle+1.0}$ | $29.5_{\blacktriangle+0.6}$ | 53.6 |
| DART-Math (Tong et al., 2024) | Llama 3 | 8B | $82.5_{\blacktriangle+5.9}$ | $45.3_{\blacktriangle+16.4}$ | 63.9 |
| MAmmoTH2-Plus (Yue et al., 2024) | Llama 3 | 8B | $84.1_{\blacktriangle+7.5}$ | $42.8_{\blacktriangle+13.9}$ | 63.5 |
| Llama 3.1-Instruct (Team, 2024) | Llama 3 | 8B | $84.5_{\blacktriangle+7.9}$ | $51.9_{\blacktriangle+23.0}$ | 68.2 |
| JiuZhang3.0 (Zhou et al., 2024) | Llama 3 | 8B | $88.6_{\blacktriangle+12.0}$ | $51.0_{\blacktriangle+22.1}$ | 69.8 |
| WizardMath-Llama (Luo et al., 2025) | Llama 3 | 8B | $90.3_{\blacktriangle+13.7}$ | $58.8_{\blacktriangle+29.9}$ | 74.6 |
| **EORM (Ours)** | **Llama 3** | 8B | $\mathbf{90.7}_{\blacktriangle+14.1}$ | $\mathbf{63.7}_{\blacktriangle+34.8}$ | **77.2** |
| Qwen2.5 7B (Yang et al., 2024) | Qwen 2.5 | 7B | $89.5_{\blacktriangle+0.0}$ | $63.4_{\blacktriangle+0.0}$ | 76.5 |
| **EORM (Ours)** | **Qwen 2.5** | 7B | $\mathbf{92.8}_{\blacktriangle+3.3}$ | $\mathbf{65.8}_{\blacktriangle+2.4}$ | **79.3** |

Table 2: **Performance comparison on math reasoning benchmarks.** We evaluate EORM on GSM8K and MATH using five LLM structures. Note that baselines such as WizardMath and Meta-Math involve extensive specialist fine-tuning. This table is intended to provide a contextual comparison, demonstrating that EORM applied post-hoc to a base model can achieve performance competitive with these computationally intensive methods.

2023), DeepSeekMath-7B (Shao et al., 2024), Llama 3 8B (Grattafiori et al., 2024), Qwen 2.5 7B (Yang et al., 2024), and Llama 2 7B (Touvron et al., 2023).

EORM was trained using a pairwise Bradley-Terry loss function (Liu, 2009). The training dataset was constructed from Chain-of-Thought (CoT) solutions for problems in the in-domain GSM8k and MATH training splits. These solutions were generated using all five aforementioned LLMs. For each training problem, we generated $n = 256$ CoT candidates with a temperature of $0.7$ and a top-$p$ (nucleus) sampling of $0.9$ (corrected from "sampling probability"), ensuring all attempts were included, regardless of their correctness. Each training instance comprised the original question, a generated solution, and a label indicating whether the solution was correct ($y^+$) or incorrect ($y^-$). EORM learned from these pairs to assign lower energy scores to preferred solutions by interpreting classifier logits as negative energies (Grathwohl et al., 2019; Liu et al., 2020). The GPT-2 tokenizer (Radford et al., 2019) was employed for the energy-based tokenizer.

To provide a rigorous comparison, our "ORM" baseline represents a standard, large-scale reward model setup. It utilizes the Llama 3 8B base model equipped with a reward head and is trained using the standard pairwise ranking loss (identical to the ranking objective used in WizardMath). This

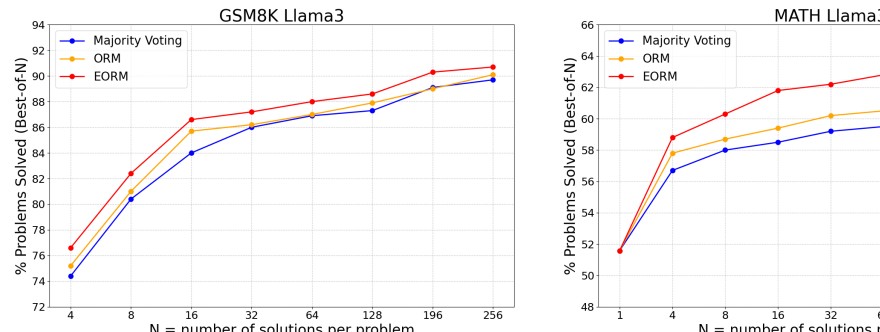

Figure 3: **EORM performance with varying samples per question.** We conduct experiments to show how the number of samples influences the problem-solving rate, using accuracy as the metric. The results indicate that model performance improves as the number of samples increases.

allows us to compare our 55M parameter EORM directly against a standard reward model that is over 145x larger.

For evaluation, we generated Chain-of-Thought (CoT) candidates across all models using a temperature of $0.7$ and a top-$p$ sampling of $0.9$. Specifically, $n = 256$ candidates were generated for each in-distribution problem, while $n = 64$ candidates were generated for each out-of-distribution problem. Subsequently, EORM was used post-hoc to select the candidate with the lowest energy. The final answer accuracy is reported in the two subsections that follow.

## 4.2 IN DISTRIBUTION LEARNING

Our primary evaluation focused on EORM's performance within the domains it was trained on, namely the GSM8k and MATH test sets. The quantitative outcomes, detailed in Table 2, consistently show that employing EORM as a post-hoc reranker leads to substantial improvements in final answer accuracy compared to the baseline performance of the underlying LLMs generating the candidate solutions. By effectively identifying and selecting the most plausible reasoning path from the generated set, EORM significantly enhances problem-solving capabilities. For instance, when integrated with Llama 3 8B and reranking $n = 256$ candidate solutions, EORM achieved high accuracy levels of 90.7% on GSM8k and 63.7% on MATH, demonstrating its ability to leverage multiple reasoning attempts effectively. The relationship between the number of candidate samples ($n$) and the resulting accuracy improvement is visualized in Figure 3, illustrating how performance generally scales with more samples but strong gains are often achievable with moderate sampling budgets. Beyond quantitative accuracy, qualitative analyses provide deeper insights into EORM's function. In addition, to assess the upper bound of reranking, we analyzed how often a correct answer exists in the candidate pool. At $n = 256$, the oracle accuracy for Llama 3 8B is 94.6% on GSM8k and 78.8% on MATH. EORM achieves 90.7% and 63.7% respectively, effectively recovering 95.9% (GSM8k) and 80.8% (MATH) of the theoretically possible performance.

## 4.3 OUT OF DISTRIBUTION LEARNING

| Model | AIME 2024 | AMC | SAT Math | Gaokao Math | Avg |
|---|---|---|---|---|---|
| Llama-3 8B | 3.3 ▲+0.0 | 19.3 ▲+0.0 | 77.3▲+0.0 | 48.7▲+0.0 | 37.2 |
| + TTRL (Zuo et al., 2025) | 3.3▲+0.0 | 32.5▲+13.2 | 89.1▲+11.8 | 61.0▲+12.3 | 46.5 |
| + DART-MATH (Prop2Diff) (Tong et al., 2024) | 6.7▲+3.4 | 26.5▲+7.2 | **90.5**▲+13.2 | 67.6▲+18.9 | 47.8 |
| **+ EORM Ours** | **10.0**▲+6.7 | **28.9**▲+9.6 | **90.5**▲+13.2 | **70.3**▲+21.6 | **49.9** |
| Qwen-2.5 7B | 16.7▲+0.0 | 53.0▲+0.0 | 91.4▲+0.0 | 83.3▲+0.0 | 61.1 |
| + TTRL (Zuo et al., 2025) | **43.3**▲+26.6 | 67.5▲+14.5 | 95.0▲+3.6 | 88.2▲+4.9 | 73.5 |
| **+ EORM Ours** | **43.3**▲+26.6 | **68.7**▲+15.7 | **96.4**▲+5.0 | **88.9**▲+5.6 | **74.3** |

Table 3: **Comparison of EORM with other reasoning methods.** We evaluate EORM against two reasoning baselines, TTRL and DART-MATH, using accuracy as the metric across four mathematical datasets. Each method uses 64 samples per question for a fair comparison. EORM consistently achieves the highest performance, with the best Accuracy values highlighted in bold.

A critical aspect of evaluating any reasoning model is its ability to generalize to new, unseen problems and potentially different reasoning styles. To rigorously test this, we assessed the performance of EORM (trained exclusively on GSM8k and MATH data) on a suite of OOD benchmarks known for their difficulty: AIME 2024, AMC, and AGIEval's SAT Math and Gaokao Math subsets (Zhong et al., 2023). These tasks often require more complex mathematical insights or different problem-solving strategies than those predominant in the training datasets. For this phase, evaluations primarily utilized Llama-3 8B and Qwen-2.5 7B as the base models, generating $n = 64$ candidate solutions for each OOD problem instance. The results, documented in Table 5, demonstrate EORM's strong generalization capabilities. When applied to the Llama-3 base model, EORM generally yielded superior performance compared to alternative baseline reranking methods like TTRL (Zuo et al., 2025) and MathWizard (Luo et al., 2025) across the majority of the evaluated OOD datasets under the same experimental setup. Impressively, EORM achieved significant scores on highly challenging benchmarks such as AIME 2024 (reaching 10.0%) and the AGIE Gaokao Math problems (achieving 70.3%). Additionally the Qwen-2.5 base model also find similar finding, and provides further evidence that EORM is not merely overfitting to patterns specific to GSM8k and MATH. Instead, it appears to learn more fundamental, transferable principles related to the structure and validity of mathematical reasoning, allowing it to effectively discern higher-quality solutions even when faced with novel problem types and domains.

## 4.4 GENERALIZATION CAPABILITIES

To further assess EORM's robustness, we analyze its generalization capabilities when encountering outputs from LLMs not seen during training. This is crucial for real-world applicability where new models are constantly emerging. We test two scenarios: generalization to unseen models on in-distribution tasks, and a more challenging test of generalization to unseen models on out-of-distribution tasks.

| Model | Base Model | Params | GSM8k | MATH |
|---|---|---|---|---|
| Mistral-v0.1 (Jiang et al., 2023) | - | 7B | $42.9_{\blacktriangle+0.0}$ | $12.9_{\blacktriangle+0.0}$ |
| EORM Generalize | Mistral-v0.1 | 7B | $76.0_{\blacktriangle+33.1}$ | $23.0_{\blacktriangle+10.1}$ |
| EORM | Mistral-v0.1 | 7B | $91.0_{\blacktriangle+48.1}$ | $48.8_{\blacktriangle+35.9}$ |
| Llama2 (Touvron et al., 2023) | - | 7B | $14.6_{\blacktriangle+0.0}$ | $2.5_{\blacktriangle+0.0}$ |
| EORM Generalize | Llama 2 | 7B | $38.5_{\blacktriangle+23.9}$ | $7.8_{\blacktriangle+5.3}$ |
| EORM | Llama 2 | 7B | $75.6_{\blacktriangle+61.0}$ | $21.8_{\blacktriangle+19.3}$ |
| DeepSeekMath-Base (Shao et al., 2024) | - | 7B | $64.2_{\blacktriangle+0.0}$ | $36.2_{\blacktriangle+0.0}$ |
| EORM Generalize | DeepSeekMath | 7B | $81.6_{\blacktriangle+17.4}$ | $41.7_{\blacktriangle+5.5}$ |
| EORM | DeepSeekMath | 7B | $84.2_{\blacktriangle+20.0}$ | $58.7_{\blacktriangle+22.5}$ |
| Llama3 (Team, 2024) | - | 8B | $76.6_{\blacktriangle+0.0}$ | $28.9_{\blacktriangle+0.0}$ |
| EORM Generalize | Llama 3 | 8B | $76.6_{\blacktriangle+0.0}$ | $51.6_{\blacktriangle+31.5}$ |
| EORM | Llama 3 | 8B | $90.7_{\blacktriangle+47.8}$ | $63.7_{\blacktriangle+43.6}$ |
| Qwen2.5 (Yang et al., 2024) | - | 7B | $89.5_{\blacktriangle+0.0}$ | $63.4_{\blacktriangle+0.0}$ |
| EORM Generalize | Qwen 2.5 | 7B | $90.2_{\blacktriangle+0.7}$ | $63.8_{\blacktriangle+0.4}$ |
| EORM | Qwen 2.5 | 7B | $92.8_{\blacktriangle+3.3}$ | $65.8_{\blacktriangle+2.4}$ |

Table 4: **Performance on in-distribution benchmarks (GSM8k, MATH) when EORM is trained without exposure to the target model's outputs.** EORM Generalize is trained on CoT data from four LLMs and tested on the fifth (target) LLM. EORM (from Table 2) is trained on data from all five LLMs. Base performance of the target LLM is also provided. Accuracy is reported.

**Generalization to Unseen Models on In-Distribution Tasks.** We employ a leave-one-out training strategy where a variant of EORM, termed "EORM Generalize," is trained on CoT data from four of the five LLMs and tested on the held-out model. The results, detailed in Table 4, show that this variant consistently improves accuracy over the base performance. For instance, when Mistral-v0.1 is the held-out model, its base accuracy on GSM8k is 42.9%. "EORM Generalize" elevates this to 76.0%, while the standard EORM (trained on all models) reaches 91.0%. The significant uplift from "EORM Generalize" demonstrates that our model learns generalizable features of correct

reasoning, rather than merely overfitting to the stylistic patterns of the models in its training set. The performance gap between the standard EORM and "EORM Generalize" is expected, as the latter lacks exposure to the target LLM's specific error patterns and stylistic nuances. However, the crucial finding is that "EORM Generalize" still provides a substantial boost, indicating that it captures fundamental principles of logical consistency and procedural correctness. This ability to transcend the idiosyncrasies of individual LLMs confirms that EORM is learning the underlying structure of valid mathematical arguments.

| Model | AIME 2024 | AMC | SAT Math | Gaokao Math | Avg |
|---|---|---|---|---|---|
| Llama-3 8B (Team, 2024) | $3.3_{\blacktriangle+0.0}$ | $19.3_{\blacktriangle+0.0}$ | $77.3_{\blacktriangle+0.0}$ | $48.7_{\blacktriangle+0.0}$ | 37.2 |
| EORM Generalized | $6.7_{\blacktriangle+3.4}$ | $20.5_{\blacktriangle+1.2}$ | $82.3_{\blacktriangle+5.0}$ | $52.6_{\blacktriangle+3.9}$ | 40.5 |
| EORM | $10.0_{\blacktriangle+6.7}$ | $28.9_{\blacktriangle+9.6}$ | $90.5_{\blacktriangle+13.2}$ | $70.3_{\blacktriangle+21.6}$ | 49.9 |
| Qwen-2.5 7B (Yang et al., 2024) | $16.7_{\blacktriangle+0.0}$ | $53.0_{\blacktriangle+0.0}$ | $91.4_{\blacktriangle+0.0}$ | $83.3_{\blacktriangle+0.0}$ | 61.1 |
| EORM Generalized | $26.7_{\blacktriangle+10.0}$ | $54.2_{\blacktriangle+1.2}$ | $92.3_{\blacktriangle+0.9}$ | $84.5_{\blacktriangle+1.2}$ | 64.4 |
| EORM | $43.3_{\blacktriangle+26.6}$ | $68.7_{\blacktriangle+15.7}$ | $96.4_{\blacktriangle+5.0}$ | $88.9_{\blacktriangle+5.6}$ | 74.3 |

Table 5: **Comparison of EORM with against baseline for out-of distribution problem and answer.** We gray shaded the generalized result.

**Generalization to Unseen Models and Tasks (Out-of-Distribution).**  We test a more demanding scenario where the "EORM Generalize" variant is evaluated on challenging OOD benchmarks using CoT solutions from the held-out LLM. This setup assesses EORM's ability to transfer learned principles to both novel tasks and unfamiliar model outputs simultaneously. The results in Table 5 show that even under this compounded challenge, our model provides tangible improvements. For example, with Llama 3 8B as the held-out model, "EORM Generalize" raises the average OOD accuracy from a base of 37.2% to 40.5%. This suggests EORM internalizes fundamental and transferable heuristics about the structure and coherence of valid reasoning that apply broadly. As anticipated, the standard EORM—which benefited from seeing the target model's in-distribution outputs during training—achieves superior OOD performance. Nevertheless, the fact that "EORM Generalize" still delivers tangible improvements in this dual-unseen scenario is a strong testament to its robustness. This finding highlights that the energy functions learned by EORM are not merely memorizing solution patterns but are instead capturing more abstract, transferable markers of high-quality reasoning applicable across diverse problem domains.

## 5 ABLATION STUDIES

To validate the contributions of key components within the EORM framework, we conducted a series of ablation studies. We investigated four primary aspects: the choice of model architecture, the impact of the tokenizer, the training objective, and data efficiency.

**Architecture and Pooling.**  We assessed the importance of the Transformer architecture by comparing our standard EORM against a simpler baseline where the Transformer encoder was replaced with a standard Multi-Layer Perceptron (MLP) verifier. To handle variable-length CoT sequences for the MLP baseline, we applied average (mean) pooling across the token embeddings to produce a fixed-size input vector. As shown in Table 4, the Transformer-

| Architecture | GSM8k | MATH |
|---|---|---|
| RNN Verifier | $78.3_{\blacktriangle+0.0}$ | $46.2_{\blacktriangle+0.0}$ |
| MLP Verifier | $82.6_{\blacktriangle+4.3}$ | $52.1_{\blacktriangle+5.9}$ |
| **Transformer (EORM)** | $\mathbf{90.7}_{\blacktriangle+12.4}$ | $\mathbf{63.7}_{\blacktriangle+17.5}$ |

Figure 4: **Impact of Architecture.** The Transformer backbone significantly outperforms MLP and RNN baselines due to better modeling of long-range dependencies.

based EORM significantly outperforms the MLP variant, achieving accuracies of 90.7% on GSM8k and 63.7% on MATH, compared to the MLP's 82.6% and 52.1%, respectively. To further validate the architecture, we expanded our comparison to include Recurrent Neural Networks (RNNs). The RNN verifier significantly underperforms (78.3% on GSM8k), confirming that the Transformer's ability to model long-range dependencies is superior to recurrent approaches for this task.

**Tokenizer Sensitivity.** Next, we examined EORM's sensitivity to the tokenizer by training it with two different tokenizers: a universal GPT-2 tokenizer and the native tokenizer from the Llama 3 model family. As presented in Table 5, the choice of tokenizer has a negligible impact on performance. The model achieves nearly identical scores on both GSM8k (86.6% vs. 86.8%) and MATH (61.8% for both). This robustness suggests that EORM learns fundamental, transferable features of valid reasoning that are not dependent on specific tokenization schemes, underscoring its generalizability. Consequently, a single EORM instance can be effectively deployed to verify outputs from diverse LLMs without the computational overhead of aligning tokenizers or retraining for specific model vocabularies.

| Tokenizer | GSM8k | MATH |
|---|---|---|
| Llama 3 (Native) | 86.8▲+0.0 | 61.8▲+0.0 |
| **GPT-2 (Universal)** | **86.6▼-0.2** | **61.8▲+0.0** |

Figure 5: **Impact of Tokenizer.** EORM is robust to tokenization choice, performing equally well with a universal tokenizer versus a model-native one. This independence simplifies deployment across heterogeneous model ecosystems without requiring tokenizer retraining.

**Loss Function Analysis.** We investigated the impact of the training objective by comparing EORM against alternative loss functions. As shown in Table 6, the Bradley-Terry ranking loss outperforms both Binary Cross-Entropy (BCE) and InfoNCE losses, particularly on the more complex MATH dataset. This confirms that the pairwise ranking objective is better suited for discriminating between subtle reasoning differences than standard classification or contrastive losses. By explicitly maximizing the energy margin between correct and incorrect pairs, the model learns to prioritize relative solution quality over absolute probability, which is critical for effective Best-of-N reranking.

| Loss Function | GSM8k | MATH |
|---|---|---|
| BCE Loss | 89.8▲+0.0 | 59.1▲+0.0 |
| InfoNCE Loss | 90.3▲+0.5 | 62.5▲+3.4 |
| **Bradley-Terry (EORM)** | **90.7▲+0.9** | **63.7▲+4.6** |

Figure 6: **Impact of Training Objective.** The pairwise Bradley-Terry ranking loss consistently outperforms standard classification (BCE) and contrastive (InfoNCE) losses.

**Data Efficiency.** Finally, we evaluated EORM's performance when trained on a subset (224k samples, $\approx 1.5\%$ of full data). As shown in Table 7, on the easier GSM8k benchmark, the model recovers 99% of its peak performance (89.9% accuracy), indicating high data efficiency for standard tasks. However, the full 15M dataset provides a substantial 12% boost on the harder MATH benchmark (51.8% vs 63.7%), suggesting large-scale data is essential for learning complex reasoning patterns. Increasing the subset size to 938k yields a significant intermediate gain on MATH (58.3%), confirming that performance scales continuously with data volume for challenging domains. This allows practitioners to trade off between training cost and peak performance: a small dataset suffices for general competence, while the massive corpus is necessary to maximize capability on intricate problems.

| Dataset Size | GSM8k | MATH |
|---|---|---|
| Subset (224k) | 89.9▲+0.0 | 51.8▲+0.0 |
| Subset (938k) | 90.3▲+0.4 | 58.3▲+6.5 |
| **Full Dataset (15M)** | **90.7▲+0.8** | **63.7▲+11.9** |

Figure 7: **Data Efficiency.** EORM learns simple tasks (GSM8k) with very little data, but large-scale data is essential for complex tasks (MATH).

## 6 CONCLUSION

We introduced EORM, a lightweight energy-based model for post-hoc verification of Chain-of-Thought reasoning. Trained solely on outcome labels, EORM efficiently reranks candidate solutions to significantly boost mathematical reasoning performance across various benchmarks. Its effectiveness, combined with a parameter count over 127x smaller than typical reward models, presents a practical path toward more dependable and accessible LLMs for complex reasoning. n future work, we plan to extend this framework to broader domains such as commonsense reasoning and explore hybrid approaches that combine the efficiency of outcome supervision with the granularity of process-level feedback.

## ETHICS STATEMENT

Our work is aimed at enhancing the reliability and efficiency of mathematical reasoning in Large Language Models (LLMs), which can reduce computational costs and foster progress in beneficial scientific and educational domains. We acknowledge, however, that the training data generation requires significant computing resources and that any technology for verifying and refining LLM outputs is potentially dual-use. For instance, it could be misused to select more plausible outputs for malicious applications. The performance of EORM is also dependent on the distribution of the training data (GSM8k, MATH), which could introduce biases if not carefully considered for broader applications. We therefore advocate for the responsible development and deployment of LLM verification models and encourage further research into their safety, fairness, and transparency to mitigate these risks.

## REPRODUCIBILITY STATEMENT

To ensure the reproducibility of our research, this paper provides a detailed account of our methodology and experimental setup. The core components of our Energy Outcome Reward Model (EORM), including the Transformer-based architecture and pairwise Bradley-Terry training objective, are described in Section 3, with the training process detailed in Algorithm 1. Our complete experimental protocol, including the LLM backbones, benchmarks, and evaluation configurations, is presented in Section 4. All hyperparameters and architectural choices are specified in the main text and Appendix D.3. We will make the source code, training scripts, and our trained EORM models publicly available upon acceptance to facilitate full verification of our results.

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

# A ALGORITHM

---

**Algorithm 1** Training EORM using Pairwise Bradley-Terry Loss

---

      **Require:**Labeled groups $\{\mathcal{Y}_n\}_{n=1}^{N}$; learning rate $\eta$; number of epochs $E$ Initialize EBM parameters $\theta$ **for** epoch = 1 to $E$ **do**

2:    **for** each group $\mathcal{Y}_n$ in the training set **do**

4:       $\mathcal{Y}_+ \leftarrow \{\text{positives in } \mathcal{Y}_n\}, \quad \mathcal{Y}_- \leftarrow \{\text{negatives in } \mathcal{Y}_n\}$

5:       **if** $\mathcal{Y}_+$ and $\mathcal{Y}_-$ are both non-empty **then**

6:         Compute energies $E_\theta(y)$ for all $y \in \mathcal{Y}_+ \cup \mathcal{Y}_-$

7:         $\mathcal{L}_n \leftarrow \dfrac{1}{|\mathcal{Y}_+||\mathcal{Y}_-|} \sum_{y_+ \in \mathcal{Y}_+} \sum_{y_- \in \mathcal{Y}_-} \log\Big(1 + \exp\big(E_\theta(y_+) - E_\theta(y_-)\big)\Big)$

8:         $\theta \leftarrow \theta - \eta \nabla_\theta \mathcal{L}_n$

9:       **end if**

10:    **end for**

11: **end for**

---

# B EXTENDED RELATED WORK

This section provides a more detailed discussion of research areas related to our work on the Energy Outcome Reward Model (EORM), covering Chain-of-Thought reasoning, techniques for verifying and reranking LLM outputs, and Energy-Based Models.

## B.1 CHAIN-OF-THOUGHT AND MULTI-STEP REASONING

Large Language Models (LLMs) (Bai et al., 2023; Touvron et al., 2023; OpenAI, 2023; Guo et al., 2025) have demonstrated remarkable capabilities, particularly when prompted to generate step-by-step solutions. Chain-of-Thought (CoT) prompting (Wei et al., 2022; Kojima et al., 2022) has become a cornerstone technique for eliciting complex multi-step reasoning from LLMs. This approach encourages models to "think aloud," breaking down problems into manageable intermediate steps. Various refinements to CoT have been proposed, such as Least-to-Most prompting (Zhou et al., 2022), which guides the model through progressively more complex steps, and Tree-of-Thoughts (ToT) (Yao et al., 2023; Zhang et al., 2023), which explores multiple reasoning paths in a structured manner. Retrieval-augmented CoT methods (Pan et al., 2025; Rakin et al., 2024; Schick et al., 2023; Yao et al., 2022; Press et al., 2022; Khattab et al., 2022) further enhance reasoning by allowing models to consult external knowledge sources or tools. Despite these advancements, CoT outputs are not infallible; a single incorrect step can derail the entire reasoning process (Tong et al., 2024). This has led to the development of post-hoc processing techniques. Among the most prominent is self-consistency (Wang et al., 2023), which samples multiple CoT outputs and selects the final answer via majority vote. While effective, self-consistency and similar ensemble methods often require generating a large number of candidate solutions, leading to substantial computational overhead (Wu et al., 2024). Our work seeks to mitigate this cost by providing a more efficient mechanism for identifying high-quality CoT solutions.

## B.2 RERANKING AND VERIFICATION OF LLM OUTPUTS

Given the variability in the quality of CoT paths, a subsequent verification or reranking stage is often crucial for improving the final answer accuracy (Wu et al., 2024; Jacovi et al., 2024; Li et al., 2023; Lin et al., 2025). Researchers have explored various strategies for this purpose. Some approaches focus on training separate verifier models to score the correctness of generated solutions or even individual reasoning steps (Cobbe et al., 2023; Khalifa et al., 2023; Jiang et al., 2024). For example, DI-VeRSe (Li et al., 2023) trains a model to perform step-aware verification. Other methods adopt a learning-to-rank perspective (Liu, 2009; Deng et al., 2020a), training models to compare and order candidate solutions based on their perceived quality using pairwise or listwise objectives. While these verification and reranking techniques can be effective, many existing methods either rely on large, specialized verifier models that add significant computational load or employ complex, computationally expensive decoding strategies. The goal of EORM is to provide a lightweight

yet powerful verifier that can efficiently identify correct CoT solutions without imposing such heavy overhead. Techniques like Chain-of-Actions (Pan et al., 2025) and Search-in-the-Chain (Xu et al., 2023) also introduce verification but often focus on different aspects or involve more intricate integration with the generation process. EORM distinguishes itself by its post-hoc applicability and its foundation in energy-based principles for efficient scoring.

## B.3 ENERGY-BASED MODELS (EBMS)

Energy-Based Models (EBMs) provide a flexible and powerful framework for modeling complex data distributions by assigning a scalar "energy" value to each input configuration (LeCun et al., 2006; Du & Mordatch, 2019). Lower energy values typically correspond to more plausible or desirable configurations. EBMs have found successful applications in diverse domains, including computer vision (Grathwohl et al., 2019), generative modeling (Song et al., 2021; Du & Mordatch, 2019), out-of-distribution detection (Liu et al., 2020), and ranking tasks (Grathwohl et al., 2019; Liu, 2009). A key characteristic of EBMs is that they do not necessarily require explicit normalization of the probability distribution (i.e., computation of the partition function), which is often intractable for high-dimensional or complex output spaces (Karakida et al., 2016; **?**). This property makes EBMs particularly well-suited for ranking scenarios, where the primary goal is to compare the relative "goodness" of different candidates rather than to compute their absolute probabilities. This is directly applicable to CoT reranking, as we aim to identify the best solution among several alternatives. A pivotal insight, leveraged by our work, is that the logits produced by standard discriminative classifiers can be interpreted as negative unnormalized log-probabilities, or effectively, as negative energies (Grathwohl et al., 2019). By adopting this perspective, an EBM can be trained using objectives similar to those used for classifiers, without the need for complex sampling procedures often associated with training generative EBMs. EORM builds on this by using a pairwise Bradley-Terry loss (Liu, 2009), a common technique in learning-to-rank, to train the energy function to distinguish between correct and incorrect CoT solutions. Our work thus extends the application of EBM principles to the domain of multi-step reasoning in LLMs, offering an efficient and theoretically grounded alternative to existing verification methods.

## C THEORETICAL DETAILS

This appendix furnishes comprehensive definitions, rigorous proofs, and supplementary remarks that underpin the theoretical concepts and analyses presented throughout the main text of the paper.

### C.1 PRELIMINARIES: FOUNDATIONS OF ENERGY-BASED MODELS

We begin by establishing the fundamental concepts of Energy-Based Models (EBMs) that form the basis of our proposed EORM.

**Definition C.1** (Energy Function). *An* energy function $E_\theta : \mathcal{Y} \to \mathbb{R}$ *is a function, parameterized by $\theta$, that assigns a scalar value $E_\theta(y)$ (its energy) to each possible configuration or input $y$ from a space $\mathcal{Y}$. By convention, lower energy values typically correspond to more desirable or probable configurations. In the* EORM *framework, $\mathcal{Y}$ denotes the space of possible Chain-of-Thought (CoT) text sequences, which are characterized by their variable length and complex linguistic and logical structures.*

**Definition C.2** (Boltzmann Distribution). *Given an energy function $E_\theta(y)$, the corresponding* Boltzmann *(or Gibbs) distribution $p_\theta(y)$ assigns a probability (or probability density for continuous $\mathcal{Y}$) to each configuration $y \in \mathcal{Y}$ as:*

$$p_\theta(y) = \frac{\exp\big(-E_\theta(y)\big)}{Z_\theta}, \tag{7}$$

*where $Z_\theta$ is the partition function.*

**Definition C.3** (Partition Function). *The* partition function $Z_\theta$ *serves as a normalization constant, ensuring that the Boltzmann distribution $p_\theta(y)$ integrates (or sums, for discrete $\mathcal{Y}$) to unity over the entire space $\mathcal{Y}$:*

$$Z_\theta = \int_{y' \in \mathcal{Y}} \exp\big(-E_\theta(y')\big)\, dy' \quad (\text{or} \sum_{y' \in \mathcal{Y}} \exp\big(-E_\theta(y')\big) \text{ for discrete } \mathcal{Y}). \tag{8}$$

*The computational intractability of $Z_\theta$ for complex, high-dimensional spaces, such as those involving text sequences, is a principal motivation for developing methods like* EORM *that can operate effectively without its explicit calculation.*

A key property of EBMs relevant to ranking tasks is the equivalence between energy minimization and probability maximization.

**Theorem C.1** (Energy–Probability Equivalence for Ranking). *For any finite candidate set $\mathcal{Y}_{\mathrm{cand}} \subset \mathcal{Y}$, the configuration $y^*$ that minimizes the energy function $E_\theta(y)$ over this set is also the configuration that maximizes the Boltzmann probability $p_\theta(y)$:*

$$y^* = \arg \min_{y \in \mathcal{Y}_{\mathrm{cand}}} E_\theta(y) = \arg \max_{y \in \mathcal{Y}_{\mathrm{cand}}} p_\theta(y). \tag{9}$$

*Proof of Theorem C.1 (Energy–Probability Equivalence for Ranking).* Consider a finite candidate set $\mathcal{Y}_{\mathrm{cand}}$ and the Boltzmann probability $p_\theta(y) = \frac{\exp(-E_\theta(y))}{Z_\theta}$ for $y \in \mathcal{Y}_{\mathrm{cand}}$. The partition function $Z_\theta$, whether defined globally over $\mathcal{Y}$ or locally over $\mathcal{Y}_{\mathrm{cand}}$ (if considering probabilities conditional on this set), is a positive constant with respect to the specific choice of $y$ from $\mathcal{Y}_{\mathrm{cand}}$. Consequently, maximizing $p_\theta(y)$ over $y \in \mathcal{Y}_{\mathrm{cand}}$ is equivalent to maximizing its numerator, $\exp(-E_\theta(y))$:

$$\arg \max_{y \in \mathcal{Y}_{\mathrm{cand}}} p_\theta(y) = \arg \max_{y \in \mathcal{Y}_{\mathrm{cand}}} \frac{\exp(-E_\theta(y))}{Z_\theta} = \arg \max_{y \in \mathcal{Y}_{\mathrm{cand}}} \exp(-E_\theta(y)).$$

The function $f(x) = \exp(-x)$ is strictly monotonically decreasing, as its derivative, $f'(x) = -\exp(-x)$, is always negative for $x \in \mathbb{R}$. A property of strictly monotonically decreasing functions is that maximizing $f(x)$ is equivalent to minimizing $x$. Applying this, maximizing $\exp(-E_\theta(y))$ is equivalent to minimizing its argument, $E_\theta(y)$:

$$\arg \max_{y \in \mathcal{Y}_{\mathrm{cand}}} \exp(-E_\theta(y)) = \arg \min_{y \in \mathcal{Y}_{\mathrm{cand}}} E_\theta(y).$$

Combining these steps, we conclude that $\arg \max_{y \in \mathcal{Y}_{\mathrm{cand}}} p_\theta(y) = \arg \min_{y \in \mathcal{Y}_{\mathrm{cand}}} E_\theta(y)$. This equivalence is fundamental to using energy functions for ranking and selection tasks. $\square$

## C.2 Details for EORM Architecture and Training Objective

The EORM model architecture employs a Transformer encoder (Vaswani et al., 2017) to process input CoT sequences. Each input sequence $y$, typically a concatenation of a question and a candidate CoT solution, is first tokenized. A special classification token (e.g., [CLS]) is prepended to the tokenized sequence; its final hidden state representation is conventionally used in Transformer models to capture a summary of the entire sequence. Let $\mathbf{h}_{\mathrm{CLS}}$ be the final hidden state vector from the Transformer encoder corresponding to this [CLS] token. This vector is then passed through a small Multi-Layer Perceptron (MLP) head, which includes Layer Normalization (Ba et al., 2016) for improved training stability. The MLP projects $\mathbf{h}_{\mathrm{CLS}}$ to a single scalar value, which EORM interprets as the energy $E_\theta(y)$ of the input sequence:

$$E_\theta(y) = \mathrm{MLP}\Big(\mathrm{LayerNorm}\big(\mathbf{h}_{\mathrm{CLS}}\big)\Big) \in \mathbb{R}. \tag{10}$$

The model is trained such that lower energy values $E_\theta(y)$ correspond to higher-quality (i.e., correct) CoT solutions.

For training EORM, the data is organized into groups. Each group $\mathcal{Y}_n$ pertains to a single problem instance $n$ and comprises multiple candidate CoT solutions $\{y_1, \ldots, y_k\}$ generated for that problem. Each candidate $y_i$ is accompanied by a binary label $l(y_i) \in \{0, 1\}$, where $l(y_i) = 1$ indicates a correct solution and $l(y_i) = 0$ indicates an incorrect one. Within each group $\mathcal{Y}_n$, we delineate two subsets: $\mathcal{Y}_+ = \{y \in \mathcal{Y}_n \mid l(y) = 1\}$ (correct solutions) and $\mathcal{Y}_- = \{y \in \mathcal{Y}_n \mid l(y) = 0\}$ (incorrect solutions). The optimization of EORM's parameters $\theta$ is driven by the pairwise Bradley-Terry loss (**?**). For a single group $\mathcal{Y}_n$ (assuming it is non-degenerate, i.e., $|\mathcal{Y}_+| > 0$ and $|\mathcal{Y}_-| > 0$), the loss is defined as:

$$\mathcal{L}(\theta; \mathcal{Y}_n) = \frac{1}{|\mathcal{Y}_+||\mathcal{Y}_-|} \sum_{y_+ \in \mathcal{Y}_+} \sum_{y_- \in \mathcal{Y}_-} \log\Big(1 + \exp\big(E_\theta(y_+) - E_\theta(y_-)\big)\Big). \tag{11}$$

This loss function penalizes instances where a correct solution $y_+$ is assigned a higher (or not sufficiently lower) energy than an incorrect solution $y_-$. The training procedure, outlined in Algorithm 1, aims to minimize this loss across all training groups.

## C.3 Interpreting Classifier Logits as Energies: Connection to Eorm

A pivotal insight that connects discriminative machine learning models to EBMs is the interpretation of classifier outputs (logits) as quantities related to energy (Grathwohl et al., 2019; Liu et al., 2020). This subsection elaborates on this connection and situates EORM's methodology within this context.

**Definition C.4** (Classifier Logits and Softmax Probability). *For a $K$-class discriminative classifier, let $f_\theta(y) = [f_1(y), \ldots, f_K(y)] \in \mathbb{R}^K$ denote the vector of output scores (logits) for an input $y$, where $\theta$ represents the classifier's parameters. The conditional probability $P(k|y;\theta)$ of $y$ belonging to class $k$ is commonly computed using the softmax function:*

$$P(k|y;\theta) = \frac{\exp(f_k(y))}{\sum_{j=1}^{K} \exp(f_j(y))}. \tag{12}$$

**Proposition C.1** (Implicit Energy Functions from Classifier Logits). *A $K$-class classifier, as described by its logits $f_\theta(y)$ and softmax probabilities $P(k|y;\theta)$ (Definition C.4), can be understood as implicitly defining $K$ class-conditional energy functions $E_k(y;\theta) = -f_k(y)$ for each class $k$.*

*1. Using these energy functions, the conditional probability $P(k|y;\theta)$ can be expressed in an explicitly energy-based form:*

$$P(k|y;\theta) = \frac{\exp(-E_k(y;\theta))}{\sum_{j=1}^{K} \exp(-E_j(y;\theta))}. \tag{13}$$

*2. Furthermore, a joint probability distribution $P(y, k; \theta)$ over inputs and classes can be consistently defined within an EBM framework. If we posit a joint energy $E(y, k; \theta) = E_k(y;\theta) + E_0(y;\theta)$, where $E_0(y;\theta)$ is an energy function associated with the input $y$ itself (representing the marginal energy of $y$), then:*

$$P(y, k; \theta) = \frac{\exp\big(-(E_k(y;\theta) + E_0(y;\theta))\big)}{Z'_\theta}, \tag{14}$$

*where $Z'_\theta = \sum_{y' \in \mathcal{Y}} \sum_{l=1}^{K} \exp\big(-(E_l(y';\theta) + E_0(y';\theta))\big)$ is the global partition function. This joint distribution correctly recovers the classifier's original conditional probabilities $P(k|y;\theta)$.*

*Proof of Proposition C.1.* :
Part 1: Derivation of the energy-based form for $P(k|y;\theta)$.

We begin with the standard softmax definition for $P(k|y;\theta)$:

$$P(k|y;\theta) = \frac{\exp(f_k(y))}{\sum_{j=1}^{K} \exp(f_j(y))}.$$

By defining the class-conditional energy $E_k(y;\theta) = -f_k(y)$, it follows that the logit $f_k(y) = -E_k(y;\theta)$. Substituting $f_k(y) = -E_k(y;\theta)$ into the numerator yields $\exp(-E_k(y;\theta))$. Similarly, substituting into each term of the sum in the denominator yields $\sum_{j=1}^{K} \exp(-E_j(y;\theta))$. Thus, the conditional probability $P(k|y;\theta)$ becomes:

$$P(k|y;\theta) = \frac{\exp(-E_k(y;\theta))}{\sum_{j=1}^{K} \exp(-E_j(y;\theta))},$$

which is Eq. (13). The denominator, $\sum_{j=1}^{K} \exp(-E_j(y;\theta))$, serves as a local (input-dependent) partition function, often denoted $Z(y;\theta)$, for the conditional distribution $P(\cdot|y;\theta)$.

Part 2: Consistent definition of the joint probability $P(y, k; \theta)$.

Let us define a joint energy function for the pair $(y, k)$ as $E(y, k; \theta) = E_k(y;\theta) + E_0(y;\theta)$. Here, $E_k(y;\theta) = -f_k(y)$ are the class-conditional energies derived from classifier logits, and $E_0(y;\theta)$ represents an energy associated with the input $y$, effectively capturing the energy of $y$ under its

marginal distribution. The joint probability $P(y, k; \theta)$ is then given by the Boltzmann distribution corresponding to this joint energy:

$$P(y, k; \theta) = \frac{\exp(-E(y, k; \theta))}{Z'_\theta} = \frac{\exp\big(-(E_k(y; \theta) + E_0(y; \theta))\big)}{Z'_\theta},$$

where $Z'_\theta = \sum_{y' \in \mathcal{Y}} \sum_{l=1}^{K} \exp\big(-(E_l(y'; \theta) + E_0(y'; \theta))\big)$ is the global partition function, summing over all possible inputs $y'$ and all classes $l$.

To confirm consistency, we derive the conditional probability $P(k|y; \theta)$ from this joint distribution using the fundamental relation $P(k|y; \theta) = \frac{P(y, k; \theta)}{P(y; \theta)}$. First, the marginal probability of $y$, $P(y; \theta)$, is obtained by summing (or marginalizing) the joint probability $P(y, l; \theta)$ over all classes $l$:

$$\begin{aligned}
P(y; \theta) &= \sum_{l=1}^{K} P(y, l; \theta) \\
&= \sum_{l=1}^{K} \frac{\exp\big(-(E_l(y; \theta) + E_0(y; \theta))\big)}{Z'_\theta} \\
&= \frac{\exp(-E_0(y; \theta))}{Z'_\theta} \sum_{l=1}^{K} \exp(-E_l(y; \theta)).
\end{aligned}$$

Now, we compute the conditional probability $P(k|y; \theta)$:

$$\begin{aligned}
P(k|y; \theta) &= \frac{P(y, k; \theta)}{P(y; \theta)} \\
&= \frac{\frac{\exp\big(-(E_k(y;\theta) + E_0(y;\theta))\big)}{Z'_\theta}}{\frac{\exp(-E_0(y;\theta))}{Z'_\theta} \sum_{l=1}^{K} \exp(-E_l(y; \theta))} \\
&= \frac{\exp\big(-(E_k(y; \theta) + E_0(y; \theta))\big)}{\exp(-E_0(y; \theta)) \sum_{l=1}^{K} \exp(-E_l(y; \theta))} \\
&= \frac{\exp(-E_k(y; \theta)) \exp(-E_0(y; \theta))}{\exp(-E_0(y; \theta)) \sum_{l=1}^{K} \exp(-E_l(y; \theta))} \\
&= \frac{\exp(-E_k(y; \theta))}{\sum_{l=1}^{K} \exp(-E_l(y; \theta))}.
\end{aligned}$$

This resulting expression for $P(k|y; \theta)$ is identical to the energy-based form derived in Part 1 (Eq. (13)) and, consequently, consistent with the original softmax formulation of the classifier (Definition C.4), given the definition $E_j(y; \theta) = -f_j(y)$. This demonstrates that a joint energy definition of the form $E(y, k; \theta) = E_k(y; \theta) + E_0(y; \theta)$ allows a standard classifier to be seamlessly integrated into a joint EBM framework. The specific choice of $E_0(y; \theta)$ influences the modeled marginal $P(y; \theta)$, but the conditional $P(k|y; \theta)$ remains determined by the classifier's logits. For example, if one aims to have $P(y, k) = P(k|y)P(y)$, then $E(y, k)$ can be set to $-\log P(k|y) - \log P(y)$, which implies a specific form for $E_0(y; \theta)$ related to $-\log P(y)$ and the local partition function $Z(y; \theta)$. However, the proposition's assertion of consistency holds for a general $E_0(y; \theta)$ representing the energy contribution of $y$. $\qquad\square$

## C.4 Conceptual Analysis of the Learned Energy Landscape

The energy function $E_\theta(y)$ learned by EORM effectively defines an "energy landscape" across the high-dimensional, discrete space $\mathcal{Y}$ of CoT solutions. The characteristics of this landscape are paramount to EORM's ability to discriminate effectively between high-quality, correct reasoning paths and flawed or incorrect ones. While EORM does not employ $E_\theta(y)$ for generative sampling (e.g., to create novel CoTs by navigating this landscape), an understanding of the intended structure of this landscape, as sculpted by the training objective, offers valuable insights into its operational mechanism.

**Definition C.5** (Optimal Energy Separation (Idealized Goal)). *Consider a specific problem or question q. Let $\mathcal{Y}_{C,q} \subseteq \mathcal{Y}$ denote the set of all correct CoT solutions for q, and $\mathcal{Y}_{I,q} \subseteq \mathcal{Y}$ denote the set of all incorrect CoT solutions for q. An ideally structured energy landscape, from the perspective of discriminating solutions for question q, would satisfy the condition:*

$$\sup_{y_c \in \mathcal{Y}_{C,q}} E_\theta(y_c) < \inf_{y_i \in \mathcal{Y}_{I,q}} E_\theta(y_i). \tag{15}$$

*This inequality implies the existence of an energy threshold $\tau_q$ such that all correct solutions for question q possess an energy $E_\theta(y_c) < \tau_q$, while all incorrect solutions have an energy $E_\theta(y_i) > \tau_q$. Achieving such perfect global separation for all possible CoTs is a highly stringent condition and unlikely to be fully realized in practice. However, the training process of EORM is designed to approximate this separation, particularly for the types of candidate solutions encountered in the training data.*

**Role of the Pairwise Bradley-Terry Loss in Shaping the Landscape.** The pairwise Bradley-Terry loss, as defined in Eq. (11), is instrumental in sculpting the desired energy landscape. For each pair of solutions $(y_+, y_-)$ from the same problem context, where $y_+$ is correct and $y_-$ is incorrect, the loss term is $\mathcal{L}_{pair}(y_+, y_-) = \log(1 + \exp(E_\theta(y_+) - E_\theta(y_-)))$. To understand its effect, let us define the "energy margin" for a correctly ordered pair as $\delta(y_+, y_-) = E_\theta(y_-) - E_\theta(y_+)$. The loss term can then be expressed as $\log(1 + \exp(-\delta(y_+, y_-)))$. Minimizing this loss is equivalent to maximizing the margin $\delta(y_+, y_-)$, thereby actively pushing $E_\theta(y_+)$ to be lower than $E_\theta(y_-)$. The gradient of this loss with respect to the energies (see components in Eq. (18)) illustrates this dynamic: The term $\sigma(E_\theta(y_+) - E_\theta(y_-))$ acts as a dynamic weighting factor.

- If $E_\theta(y_+) \geq E_\theta(y_-)$ (i.e., the pair is misordered or has no margin, $\delta(y_+, y_-) \leq 0$), the sigmoid term $\sigma(E_\theta(y_+) - E_\theta(y_-))$ is $\geq 0.5$. This results in a relatively strong gradient signal that pushes to decrease $E_\theta(y_+)$ and increase $E_\theta(y_-)$, effectively trying to correct the order and increase the margin.

- If $E_\theta(y_+) \ll E_\theta(y_-)$ (i.e., the pair is well-ordered with a large positive margin $\delta(y_+, y_-) \gg 0$), the sigmoid term approaches 0. The gradient signal becomes weak, as the desired ordering is already satisfied.

This adaptive mechanism concentrates the learning effort on problematic pairs, carving out low-energy regions for correct solutions relative to incorrect ones. The loss enforces relative energy orderings rather than absolute energy targets, which can make the training more robust, for example, to imbalances in the number of correct versus incorrect examples per problem.

## C.5 THEORETICAL ANALYSIS OF THE EORM TRAINING OBJECTIVE

This section details the mathematical properties of the EORM training objective, including definitions of relevant functions and formal derivations.

**Definition C.6** (Sigmoid Function). *The sigmoid function $\sigma : \mathbb{R} \to (0,1)$ is defined as:*

$$\sigma(z) = \frac{1}{1 + \exp(-z)}. \tag{16}$$

**Definition C.7** (Softplus Function). *The softplus function softplus $: \mathbb{R} \to \mathbb{R}_{>0}$ is defined as:*

$$softplus(z) = \log(1 + \exp(z)). \tag{17}$$

*A useful property is that its derivative is the sigmoid function: $\frac{d}{dz} softplus(z) = \sigma(z)$.*

**Theorem C.2** (Gradient of Pairwise Bradley-Terry Loss). *Let $\mathcal{L}(\theta; \mathcal{Y}_n)$ be the pairwise Bradley-Terry loss for a group $\mathcal{Y}_n$, as given by Eq. (11). Assuming the energy function $E_\theta(y)$ is differentiable with respect to its parameters $\theta$, the gradient of this loss is:*

$$\nabla_\theta \mathcal{L}(\theta; \mathcal{Y}_n) = \frac{1}{|\mathcal{Y}_+||\mathcal{Y}_-|} \sum_{y_+ \in \mathcal{Y}_+} \sum_{y_- \in \mathcal{Y}_-} \sigma\Big(E_\theta(y_+) - E_\theta(y_-)\Big)\Big(\nabla_\theta E_\theta(y_+) - \nabla_\theta E_\theta(y_-)\Big), \tag{18}$$

*where $\sigma(\cdot)$ is the sigmoid function (Definition C.6).*

*Proof of Theorem C.2.* The pairwise Bradley-Terry loss for a non-degenerate group $\mathcal{Y}_n$ (where $\mathcal{Y}_+$ and $\mathcal{Y}_-$ are non-empty) is:

$$\mathcal{L}(\theta; \mathcal{Y}_n) = \frac{1}{|\mathcal{Y}_+||\mathcal{Y}_-|} \sum_{y_+ \in \mathcal{Y}_+} \sum_{y_- \in \mathcal{Y}_-} \log\left(1 + \exp\left(E_\theta(y_+) - E_\theta(y_-)\right)\right).$$

Let $L_{pair}(y_+, y_-; \theta) = \log\left(1 + \exp\left(E_\theta(y_+) - E_\theta(y_-)\right)\right)$ denote the loss contribution from a single pair $(y_+, y_-)$. Using the definition of the softplus function (Definition C.7), we can write $L_{pair}(y_+, y_-; \theta) = \text{softplus}(E_\theta(y_+) - E_\theta(y_-))$. Due to the linearity of the gradient operator, we have:

$$\nabla_\theta \mathcal{L}(\theta; \mathcal{Y}_n) = \frac{1}{|\mathcal{Y}_+||\mathcal{Y}_-|} \sum_{y_+ \in \mathcal{Y}_+} \sum_{y_- \in \mathcal{Y}_-} \nabla_\theta L_{pair}(y_+, y_-; \theta).$$

To compute $\nabla_\theta L_{pair}(y_+, y_-; \theta)$, we apply the chain rule, noting that $\frac{d}{dz}\text{softplus}(z) = \sigma(z)$:

$$\begin{aligned}
\nabla_\theta L_{pair}(y_+, y_-; \theta) &= \nabla_\theta \text{softplus}\left(E_\theta(y_+) - E_\theta(y_-)\right) \\
&= \frac{d}{dz}\text{softplus}(z)\bigg|_{z = E_\theta(y_+) - E_\theta(y_-)} \cdot \nabla_\theta\left(E_\theta(y_+) - E_\theta(y_-)\right) \\
&= \sigma\left(E_\theta(y_+) - E_\theta(y_-)\right) \cdot \left(\nabla_\theta E_\theta(y_+) - \nabla_\theta E_\theta(y_-)\right).
\end{aligned}$$

Substituting this expression back into the sum for $\nabla_\theta \mathcal{L}(\theta; \mathcal{Y}_n)$ yields the statement of the theorem, Eq. (18). $\qquad\square$

**Remark C.1** (Pairwise Bradley-Terry Loss as Negative Log-Likelihood). *The pairwise Bradley-Terry loss function used in* EORM *has a strong theoretical grounding as the negative log-likelihood (NLL) of observed preferences under the Bradley-Terry probabilistic model for pairwise comparisons (**?**). This model assumes that each item $y$ possesses an underlying positive "strength" or "score," denoted $\pi_y$. The probability that item $y_i$ is preferred over item $y_j$ is then given by $P(y_i \text{ preferred over } y_j) = \frac{\pi_i}{\pi_i + \pi_j}$. In an energy-based formulation, we can define the strength of a solution $y$ as inversely related to its energy: $\pi_y = \exp(-E_\theta(y))$. A lower energy $E_\theta(y)$ thus corresponds to a higher strength $\pi_y$. Under this definition, the probability that a correct solution $y_+$ is preferred over an incorrect solution $y_-$ (which implies $E_\theta(y_+)$ should be less than $E_\theta(y_-)$) can be modeled as:*

$$\begin{aligned}
P(y_+ \text{ preferred over } y_- | \theta) &= \frac{\pi_{y_+}}{\pi_{y_+} + \pi_{y_-}} \\
&= \frac{\exp(-E_\theta(y_+))}{\exp(-E_\theta(y_+)) + \exp(-E_\theta(y_-))}.
\end{aligned}$$

*Dividing both the numerator and the denominator by $\exp(-E_\theta(y_+))$ yields:*

$$\begin{aligned}
P(y_+ \text{ preferred over } y_- | \theta) &= \frac{1}{1 + \exp(-E_\theta(y_-) + E_\theta(y_+))} \\
&= \frac{1}{1 + \exp\left(-(E_\theta(y_-) - E_\theta(y_+))\right)} \\
&= \sigma\left(E_\theta(y_-) - E_\theta(y_+)\right), \quad \text{(using the definition of } \sigma(z) \text{ from Definition C.6).}
\end{aligned}$$

*The negative log-likelihood (NLL) of observing this single preference $y_+ > y_-$ is:*

$$\begin{aligned}
NLL(y_+ > y_- | \theta) &= -\log P(y_+ \text{ preferred over } y_- | \theta) \\
&= -\log \sigma\left(E_\theta(y_-) - E_\theta(y_+)\right).
\end{aligned}$$

*Using the identity $-\log\sigma(x) = \log(1 + e^{-x}) = \text{softplus}(-x)$, we have:*

$$\begin{aligned}
NLL(y_+ > y_- | \theta) &= \log\left(1 + \exp\left(-(E_\theta(y_-) - E_\theta(y_+))\right)\right) \\
&= \log\left(1 + \exp\left(E_\theta(y_+) - E_\theta(y_-)\right)\right).
\end{aligned}$$

*This expression is precisely the loss contribution from a single pair $(y_+, y_-)$ as defined in Eq. (11). This term is also recognizable as the logistic loss or binary cross-entropy for the event of $y_+$ being*

*preferred over $y_-$, given the "logit" of this preference is $E_\theta(y_-) - E_\theta(y_+)$. Therefore, minimizing the average pairwise Bradley-Terry loss is equivalent to performing Maximum Likelihood Estimation (MLE) for the parameters $\theta$ under this probabilistic preference model, where preferences are determined by energy differences. This equivalence provides a robust theoretical justification for the chosen loss function.*

**Proposition C.2** (Unbiased Gradient Estimation from Non-Degenerate Groups). *Let $\mathcal{G}$ represent the true underlying distribution of all training groups $n$. The loss for group $n$, $\mathcal{L}(\theta; \mathcal{Y}_n)$, is defined such that $\mathcal{L}(\theta; \mathcal{Y}_n) = 0$ if group $n$ is degenerate (i.e., it lacks either positive examples, $|\mathcal{Y}_+| = 0$, or negative examples, $|\mathcal{Y}_-| = 0$). Let $\mathcal{G}^\dagger$ be the conditional distribution of groups $n$ from $\mathcal{G}$, given that they are non-degenerate. In stochastic gradient descent (SGD), if we sample a group $n \sim \mathcal{G}^\dagger$ (i.e., we exclusively process non-degenerate groups) and compute its gradient $\nabla_\theta \mathcal{L}(\theta; \mathcal{Y}_n)$ using Eq. (11), this gradient serves as an unbiased estimate of the expected gradient over* non-degenerate *groups, $\mathbb{E}_{n \sim \mathcal{G}^\dagger}[\nabla_\theta \mathcal{L}(\theta; \mathcal{Y}_n)]$. Furthermore, this expectation is proportional to the true expected gradient over* all *groups, $\mathbb{E}_{n \sim \mathcal{G}}[\nabla_\theta \mathcal{L}(\theta; \mathcal{Y}_n)]$.*

*Proof of Proposition C.2.* Let $\mathbb{E}_{\mathcal{G}}[\cdot]$ denote the expectation with respect to the distribution $\mathcal{G}$ over all groups, and $\mathbb{E}_{\mathcal{G}^\dagger}[\cdot]$ denote the expectation with respect to the conditional distribution $\mathcal{G}^\dagger$ over non-degenerate groups. Let $\mathcal{S}_{\text{all}}$ be the set of indices for all possible training groups, $\mathcal{S}_{\text{nd}}$ be the set of indices for non-degenerate groups, and $\mathcal{S}_{\text{d}}$ be the set of indices for degenerate groups, such that $\mathcal{S}_{\text{all}} = \mathcal{S}_{\text{nd}} \cup \mathcal{S}_{\text{d}}$ and $\mathcal{S}_{\text{nd}} \cap \mathcal{S}_{\text{d}} = \emptyset$.

The true expected gradient over all groups is $\mathbb{E}_{\mathcal{G}}[\nabla_\theta \mathcal{L}(\theta; \mathcal{Y}_n)]$. This can be expanded as:

$$\mathbb{E}_{\mathcal{G}}[\nabla_\theta \mathcal{L}(\theta; \mathcal{Y}_n)] = \sum_{i \in \mathcal{S}_{\text{all}}} P_{\mathcal{G}}(i) \nabla_\theta \mathcal{L}(\theta; \mathcal{Y}_i)$$

$$= \sum_{i \in \mathcal{S}_{\text{nd}}} P_{\mathcal{G}}(i) \nabla_\theta \mathcal{L}(\theta; \mathcal{Y}_i) + \sum_{i \in \mathcal{S}_{\text{d}}} P_{\mathcal{G}}(i) \nabla_\theta \mathcal{L}(\theta; \mathcal{Y}_i).$$

By our definition, for any degenerate group $i \in \mathcal{S}_{\text{d}}$, the loss $\mathcal{L}(\theta; \mathcal{Y}_i) = 0$. Consequently, its gradient $\nabla_\theta \mathcal{L}(\theta; \mathcal{Y}_i) = \mathbf{0}$ for $i \in \mathcal{S}_{\text{d}}$. Thus, the sum simplifies to:

$$\mathbb{E}_{\mathcal{G}}[\nabla_\theta \mathcal{L}(\theta; \mathcal{Y}_n)] = \sum_{i \in \mathcal{S}_{\text{nd}}} P_{\mathcal{G}}(i) \nabla_\theta \mathcal{L}(\theta; \mathcal{Y}_i).$$

Let $P(\mathcal{S}_{\text{nd}}) = \sum_{i \in \mathcal{S}_{\text{nd}}} P_{\mathcal{G}}(i)$ be the total probability of sampling a non-degenerate group. We assume $P(\mathcal{S}_{\text{nd}}) > 0$ (otherwise, no training would occur). The probability of sampling a specific non-degenerate group $i \in \mathcal{S}_{\text{nd}}$ under the conditional distribution $\mathcal{G}^\dagger$ (i.e., given that the sampled group is non-degenerate) is $P_{\mathcal{G}^\dagger}(i) = P_{\mathcal{G}}(i | i \in \mathcal{S}_{\text{nd}}) = \frac{P_{\mathcal{G}}(i)}{P(\mathcal{S}_{\text{nd}})}$. The expected gradient when sampling exclusively from non-degenerate groups is:

$$\mathbb{E}_{\mathcal{G}^\dagger}[\nabla_\theta \mathcal{L}(\theta; \mathcal{Y}_n)] = \sum_{i \in \mathcal{S}_{\text{nd}}} P_{\mathcal{G}^\dagger}(i) \nabla_\theta \mathcal{L}(\theta; \mathcal{Y}_i)$$

$$= \sum_{i \in \mathcal{S}_{\text{nd}}} \frac{P_{\mathcal{G}}(i)}{P(\mathcal{S}_{\text{nd}})} \nabla_\theta \mathcal{L}(\theta; \mathcal{Y}_i)$$

$$= \frac{1}{P(\mathcal{S}_{\text{nd}})} \sum_{i \in \mathcal{S}_{\text{nd}}} P_{\mathcal{G}}(i) \nabla_\theta \mathcal{L}(\theta; \mathcal{Y}_i).$$

Comparing the two expectations, we find:

$$\mathbb{E}_{\mathcal{G}}[\nabla_\theta \mathcal{L}(\theta; \mathcal{Y}_n)] = P(\mathcal{S}_{\text{nd}}) \cdot \mathbb{E}_{\mathcal{G}^\dagger}[\nabla_\theta \mathcal{L}(\theta; \mathcal{Y}_n)].$$

Since $P(\mathcal{S}_{\text{nd}})$ is a positive constant (typically close to 1 if degenerate groups are infrequent), the gradient $\nabla_\theta \mathcal{L}(\theta; \mathcal{Y}_n)$ for $n \sim \mathcal{G}^\dagger$ (as computed in **??**) is an unbiased estimate of $\mathbb{E}_{\mathcal{G}^\dagger}[\nabla_\theta \mathcal{L}(\theta; \mathcal{Y}_n)]$. Furthermore, this expected gradient over non-degenerate groups is directly proportional to the true expected gradient over all groups. In the context of SGD, this means that updating parameters using gradients from only non-degenerate groups will, on average, follow a direction proportional to the true full-batch gradient direction. This justifies the common practice of skipping degenerate groups during training, as it maintains an optimization path towards minimizing the true expected loss, with the effective learning rate scaled by $P(\mathcal{S}_{\text{nd}})$. $\qquad\square$

## D  IMPLEMENTATION DETAILS

This section provides key code snippets illustrating the implementation of the EORM model architecture and the training objective used in this work.

### D.1  EORM MODEL ARCHITECTURE (TRANSEBM)

The EORM model is implemented using PyTorch, leveraging a standard Transformer encoder architecture. The core components are defined in the TransEBM class, shown in Listing 1. The model takes token IDs and an attention mask as input and outputs a single scalar energy value for the sequence, derived from the final hidden state of the prepended [CLS] token. The variable representing the model's hidden dimension is referred to as dim_model.

Listing 1: PyTorch implementation of the TransEBM model.

```python
import torch
import torch.nn as nn

class TransEBM(nn.Module):
    """Lightweight Transformer-based Energy-Based Model."""
    def __init__(self,
                 vocab_size: int,
                 dim_model: int,
                 n_heads: int,
                 n_layers: int,
                 dropout: float):
        super().__init__()
        self.dim_model = dim_model
        self.emb = nn.Embedding(vocab_size, dim_model)
        # Standard Transformer Encoder Layer
        encoder_layer = nn.TransformerEncoderLayer(
            d_model=dim_model,
            nhead=n_heads,
            dim_feedforward=4 * dim_model,
            activation="gelu",
            dropout=dropout,
            batch_first=True,
            norm_first=True
        )
        # Stack of encoder layers
        self.enc = nn.TransformerEncoder(encoder_layer, num_layers=
            n_layers)
        # MLP head to project CLS representation to scalar energy
        self.head = nn.Sequential(
            nn.LayerNorm(dim_model),
            nn.Linear(dim_model, dim_model),
            nn.GELU(),
            nn.Linear(dim_model, 1)
        )

    def forward(self, ids: torch.Tensor, mask: torch.Tensor) -> torch.
        Tensor:
        """
        Args:
            ids: (B, L) token ids (long tensor)
            mask: (B, L) attention mask (1=real, 0=pad) (long tensor)
        Returns:
            energies: (B,) scalar energy for each sequence (float tensor)
        """
        # Get embeddings, apply scaling using renamed dim
        x = self.emb(ids) * (self.dim_model**0.5)
        # Create padding mask for self-attention
        # True where attention should be masked (i.e., where mask == 0)
        padding_mask = (mask == 0)
```

```
            # Pass through Transformer encoder
            x = self.enc(x, src_key_padding_mask=padding_mask)
            # Use the representation of the first token ([CLS])
            cls_representation = x[:, 0]
            # Compute scalar energy using the MLP head
            # .squeeze(-1) removes the trailing dimension of size 1
            energy = self.head(cls_representation).squeeze(-1)
            return energy
```

## D.2 PAIRWISE BRADLEY-TERRY LOSS FUNCTION

The model is trained using a pairwise Bradley-Terry objective, implemented as shown in Listing 2. This function takes the energy scores assigned by the model to a group of candidate solutions and their corresponding binary labels (1 for correct, 0 for incorrect). It computes the average loss over all pairs of correct and incorrect candidates within the group, encouraging lower energy for correct solutions.

Listing 2: PyTorch implementation of the Pairwise Bradley-Terry loss.

```
import torch
import torch.nn.functional as F

def bradley_terry_loss(energies: torch.Tensor, labels: torch.Tensor):
    """
    Calculates pairwise Bradley-Terry based loss for a group.
    Args:
        energies: (N,) tensor of energy scores for N candidates.
        labels: (N,) tensor of binary labels (1.0=correct, 0.0=incorrect)
            .
    Returns:
        Scalar loss tensor, or None if no valid pairs exist.
    """
    # Find indices of positive (correct) and negative (incorrect)
        examples
    pos_indices = torch.where(labels == 1.0)[0]
    neg_indices = torch.where(labels == 0.0)[0]

    # Handle degenerate groups (no positives or no negatives)
    if len(pos_indices) == 0 or len(neg_indices) == 0:
        return None # No pairs to compare

    # Get energies for positive and negative examples
    pos_energies = energies[pos_indices]
    neg_energies = energies[neg_indices]

    # Calculate all pairwise energy differences: E(positive) - E(negative
        )
    # Broadcasting creates a matrix of shape (num_pos, num_neg)
    energy_diffs = pos_energies.unsqueeze(1) - neg_energies.unsqueeze(0)

    # Calculate loss for each pair using softplus: log(1 + exp(diff))
    # This is equivalent to the NLL of P(positive > negative)
    loss_matrix = F.softplus(energy_diffs)

    # Return the mean loss over all pairs
    return loss_matrix.mean()
```

## D.3 HYPERPARAMETER SETTINGS

The training and evaluation of the EORM model were conducted using a specific set of hyperparameters, primarily configured via command-line arguments with defaults specified in the training script. Key architectural parameters for the EORM model (a Transformer-based EBM) and crucial settings

Table 6: **Key hyperparameters for EORM model architecture and training.**

| Category | Hyperparameter | Value |
|---|---|---|
| **EORM Model Architecture** | Embedding Dimension | 4096 |
| | Transformer Encoder Layers | 2 |
| | Attention Heads | 4 |
| | Dropout Rate | 0.2 |
| | Feed-forward Dimension | $4 \times \mathrm{d\_model}$ |
| | Activation Function | GELU |
| | Normalization Style | Pre-LN (NormFirst) |
| **Tokenizer Configuration** | Base Tokenizer | GPT-2 |
| | Max Sequence Length | 4096 |
| | CLS ID | BOS token of tokenizer |
| | PAD ID | EOS token |
| **Training Configuration** | Optimizer | AdamW |
| | Learning Rate | $1 \times 10^{-4}$ |
| | Weight Decay | 0.01 |
| | LR Scheduler | Cosine with Warmup |
| | Warmup Ratio | 0.2 (of total steps) |
| | Number of Epochs | 50 |
| | Batch Size (groups) | 1 |
| | Gradient Clipping Norm | 1.0 |
| | FP16 (AMP) | CUDA |
| | Training/Validation Split | 80% / 20% |
| | DataLoader | 2 |

for the optimization process are detailed in Table 6. The AdamW optimizer was used in conjunction with a cosine learning rate schedule including a warmup phase. Training was performed with Automatic Mixed Precision (AMP) if the '–fp16' flag was enabled and a CUDA-compatible GPU was available. The tokenizer specified by default was 'gpt2', with its vocabulary size determining the model's embedding layer dimensions. A fixed random seed (42) was used for shuffling and creating the 80/20 train-validation split from the combined training data.

## E    DATASET DETAILS

This section outlines the dataset utilized for training and evaluating our EORM model. We describe the data format, present a comparison with other notable mathematical reasoning datasets, and specify the overall size of our training corpus. The core of our dataset comprises (question, Chain-of-Thought solution, correctness label) triplets. These CoT solutions were generated by employing a suite of Large Language Models to solve problems sourced from established mathematical reasoning benchmarks, principally the training splits of GSM8k (Cobbe et al., 2021a) and MATH (Hendrycks et al., 2021), as elaborated in the main experimental setup (Section 4).

### E.1    DATA FORMAT

Each instance in our dataset is represented as a JSON object, with multiple instances typically stored in a JSON Lines ('.jsonl') file. Each JSON object contains the original mathematical problem, a generated CoT solution, and a binary label indicating whether the provided solution correctly answers the problem. An illustrative example of a single data instance is presented in Appendix E.1.



**Training Dataset Format Example**

**label:** 1

**question:**
How many vertical asymptotes does the graph of

$$y = \frac{2}{x^2 + x - 6}$$

have?

**gen_text:**

To determine the number of vertical asymptotes for the function $y = \frac{2}{x^2 + x - 6}$, we need to identify the values of $x$ that make the denominator zero, as these are the points where the function is undefined and potentially has vertical asymptotes.

1. *Factor the denominator*
   $x^2 + x - 6 = (x + 3)(x - 2)$

2. *Solve for zeros*
   $(x + 3)(x - 2) = 0 \implies x = -3 \quad \text{or} \quad x = 2$

3. *Check for holes*
   The numerator is the constant 2, which is non-zero at $x = -3$ and $x = 2$, so no factors cancel. Hence each zero of the denominator corresponds to a vertical asymptote.

Therefore, the graph has

$$\boxed{2}$$

vertical asymptotes: one at $x = -3$ and one at $x = 2$.



### E.2    COMPARISON BETWEEN OTHER DATASET

Table 7: **Comparison of the EORM training data scale and problem synthesis approach with other mathematical reasoning datasets.** Dataset sizes are approximate, in thousands (k) of samples. "Synthesis Agent" refers to the primary method or model used for generating the core problems in the dataset.

| Dataset | Synthesis Agent | Dataset Size (k) |
|---|---|---|
| WizardMath (Luo et al., 2023) | GPT-4 | 96 |
| MetaMathQA (Yu et al., 2024) | GPT-3.5 | 395 |
| MMIQC (Liu et al., 2024a) | GPT-4+GPT-3.5+Human | 2294 |
| Orca-Math (Mitra et al., 2024) | GPT-4 | 200 |
| Xwin-Math-V1.1 (Li et al., 2024) | GPT-4 | 1440 |
| KPMath-Plus (Huang et al., 2024) | GPT-4 | 1576 |
| MathScaleQA (Tang et al., 2024) | GPT-3.5+Human | 2021 |
| DART-Math | DeepSeekMath-7B-RL | 591 |
| **EORM** | **None** | **14958** |

To position our data generation efforts for training EORM, Table 7 provides a comparative overview against several other prominent datasets used in mathematical reasoning research. The table highlights the scale (approximate number of samples) and the primary "Synthesis Agent" responsible for creating the core problems or question-answer pairs in those datasets. Many existing datasets, as indicated, leverage powerful large language models (LLMs) like GPT-4 or GPT-3.5, sometimes augmented with human effort, to synthesize novel mathematical problems. In contrast, our approach for the EORM training data (labeled as "EORM" in the table) does not involve the synthesis of new problems. Instead, we focus on generating a very large corpus of Chain-of-Thought (CoT) solutions

by prompting various existing LLMs with problems from established, open benchmarks (such as GSM8k and MATH). Each generated CoT solution is then labeled for correctness based on its final answer. This strategy, reflected by "None" under "Synthesis Agent" for our EORM data, results in a distinct type of dataset geared towards learning to verify reasoning processes rather than generating new problem instances. The substantial size of our collected data, approximately 14.96 million (14958k) (question, CoT solution, label) triplets, is intended to provide a diverse and comprehensive basis for training a robust EORM verifier capable of understanding a wide range of reasoning styles and error patterns.

## F   HARDWARE RESOURCES

The computational experiments presented in this paper, encompassing both the generation of Chain-of-Thought (CoT) solutions and the training of our Energy Outcome Reward Model (EORM), utilized high-performance Graphics Processing Units (GPUs). For the initial phase of generating the CoT candidate solutions from various Large Language Models (LLMs), a range of NVIDIA GPUs was employed. This included access to NVIDIA A800 (40GB), NVIDIA A100 (with both 40GB and 80GB variants), NVIDIA RTX A6000 Ada (48GB), and NVIDIA H100 GPUs (80GB). The availability of this diverse hardware allowed for extensive data generation across multiple LLM architectures. The training of the EORM model itself was conducted on a more focused set of high-end accelerators. Specifically, we utilized configurations consisting of 2 to 4 NVIDIA H100 (80GB) GPUs. With a setup of 4 NVIDIA H100 (80GB) GPUs, the complete training process for the EORM model took approximately 36 hours.

## G   ETHICS AND SOCIETAL IMPACT

This research focuses on improving the correctness and efficiency of mathematical reasoning in LLMs, which can yield positive societal impacts such as accelerating scientific discovery, enhancing educational tools, and reducing the energy consumption of large-scale AI computations by minimizing flawed outputs. However, we recognize that a powerful and efficient reasoning verifier is a dual-use technology. In malicious hands, it could be used to automate the selection of the most effective or deceptive generated content, such as refining disinformation or generating plausible-but-malicious code. Furthermore, the performance of EORM is intrinsically tied to the dataset used for its training; any biases present in the problem sets (e.g., cultural context in word problems) could lead to the model favoring certain reasoning styles or performing inequitably across different problem domains. Our work is intended purely for beneficial applications, and we advocate for establishing strong ethical guidelines for the development and use of verifier and reward models in advanced LLM systems.

## H   THE USE OF LARGE LANGUAGE MODELS (LLMS)

Large Language Models (LLMs) are central to this research in two primary ways. First, they are the object of study; our EORM framework is designed to evaluate and improve the reasoning outputs of base models such as Llama 3, Mistral, and DeepSeekMath. Second, these same LLMs were used as a critical tool to generate the large-scale dataset of Chain-of-Thought solutions (both correct and incorrect) that EORM was trained on. For the preparation of this manuscript, our use of LLMs was strictly limited to polishing language and generating figures. All underlying research and intellectual content, including the EORM framework, its theoretical foundations, the experimental design, and the analysis of results, was completed entirely by the authors.

