# OpenReview forum: "Learning to Rank Chain-of-Thought: Using a Small Model"
_ICLR.cc/2026/Conference — ICLR 2026 Conference Desk Rejected Submission_

### Official Review · Reviewer_sf5M · 2025-10-21

**Soundness:** 2
**Presentation:** 2
**Contribution:** 3
**Rating:** 6
**Confidence:** 3

**Summary:**

The work shows that a relatively small (~55 M parameters) energy-based transformer encoder can be used as an efficient reward model for math-oriented tasks.
It provides a comparison with an established majority voting baseline, along with fine-tuned models and methods such as test-time RL.
The results show that there seems to be a strong case for the EBM reward model, showing its ability to generalize beyond the distribution of training data and model completions.
However, the work seems to underexplore several important aspects:
+ importance of Bradley-Terry loss on logits vs classification approach
+ comparison to reward models created by tuning a reward head on top of an autoregressive transformer (or maybe this is the ORM type used in Figure 3)

**Strengths:**

+ The proposed method is evaluated against a large collection of fine-tuned models (Mistral, Llama2, DeepSeekMath, Llama 3, and Qwen 2.5) with parameters ranging around 7B.
+ The work shows how to create a relatively small and powerful reward model that beats majority voting and fine-tuned models

**Weaknesses:**

Presentation:

- Figure 1: Using an incorrect solution (70) to the "daps baps" problem as an example of something that an EORM thinks is correct and marking it in green can be misleading.
Note that the correct answer is 40, as 4 * 10 daps = 7*10 yaps and 5*14 yaps = 70 yaps = 3*14 baps = 42 baps
- line 188-189 "where Zθ is the partition function, a normalization constant that ensures pθ (y) sums to unity:" -> where Zθ is a normalization constant that ensures pθ (y) sums to unity:
- 264-265 "For each training problem, we generated n = 256 CoT candidates with a temperature of 0.7 and a sampling probability of 0.9, ensuring all attempts were included, regardless of their correctness." - this is a little bit unclear way of saying that authors use nucleus sampling with temperature 0.7 and top-p=0.9
- 360-361 "Additionally, the Qwen-2.5 base model also finds similar findings."
- Table 3: lines 374-377: "Comparison of EORM with other reasoning methods. We evaluate EORM against two reasoning baselines, TTRL and MathWizard, using accuracy as the metric across four mathematical datasets.": However, only TTRL (Zuo et al., 2025), DART-MATH (Prop2Diff) (Tong et al., 2024), and EORM are presented
- line 907 has a citation error for Layer Norm (? instead of citation)

Other:
- A proper description of ORM in Figure 3 would be beneficial (how exactly does it differ from EORM)
- A proper description of how the input feeds to the MLP verifier (instead of the transformer encoder) in the ablation section would be beneficial
- minor: work does not evaluate recent "thinking" models

**Questions:**

1. For the ablation of the architecture, how is the input feed to the MLP (instead of the transformer encoder) verifier? That is how different output lengths are handled?
2. How does the ORM in Figure 3 differ from EORM?
3. What parts are really important for EORM? That is:
    + What makes it distinct from ORM trained as a small transformer model instead of a head on top of a large model?
    + How does the Bradley-Terry loss affect the performance?

---

> ### Author Response · Authors · 2025-11-21
>
> We sincerely thank you for this exceptionally careful and constructive review! Your detailed feedback on the presentation and the core methodology is invaluable. We are grateful for the list of specific points, all of which we will correct in the revised manuscript!
>
> ### Weaknesses: Presentation
>
> > **Reviewer's Comment**: 1. Figure 1: Using an incorrect solution (70) to the "daps baps" problem as an example... can be misleading... Note that the correct answer is 40...
>
> Our sincere apologies. We will replace this with a correct and clear example in the revised version. Thank you for catching this.
>
> > **Reviewer's Comment**: 2. line 188-189... 264-265... 360-361... Table 3... line 907...
>
> Thank you for this meticulous proofreading. We will correct all of these points in the revised version:
> * The definition of $Z_{\theta}$ (line 188-189).
> * "sampling probability" (line 264-265) will be corrected to "**top-p (nucleus) sampling of 0.9**."
> * The grammatical error in line 360-361.
> * The description for Table 3 will be corrected to match its contents (TTRL and DART-MATH).
> * The missing citation for Layer Norm (line 907) will be added.
>
> ### Weaknesses: Other
>
> > **Reviewer's Comment**: 1. A proper description of ORM in Figure 3 would be beneficial (how exactly does it differ from EORM)
> > **Reviewer's Comment**: 2. A proper description of how the input feeds to the MLP verifier (instead of the transformer encoder) in the ablation section would be beneficial
>
> We apologize for these omissions. These points are also raised in your questions, and we have provided detailed answers below (Q1 and Q2). In short: the **MLP verifier uses average pooling**, and the **"ORM" in Figure 3 is our 8B Llama 3 baseline** (not a 55M model).
>
> > **Reviewer's Comment**: 3. minor: work does not evaluate recent "thinking" models
>
> This is a valid point. Our focus was on general-purpose base models. Applying EORM to models that have their own internal verification or "thinking" steps is an interesting direction for future work to see if EORM can provide a complementary boost.
>
> ### Questions:
>
> > **Reviewer's Question**: 1. For the ablation of the architecture, how is the input feed to the MLP (instead of the transformer encoder) verifier? That is how different output lengths are handled?
>
> Our apologies for omitting this. To handle the variable-length input for the MLP, we applied **average pooling (mean pooling)** across the final hidden states of all tokens in the sequence. This produced a single fixed-size vector, which was then fed into the MLP. We will add this detail to the ablation section.
>
> > **Reviewer's Question**: 2. How does the ORM in Figure 3 differ from EORM?
>
> This is a key point that we will clarify. The "ORM" baseline in Figure 3 is a **standard, large-scale Llama 3 8B model** with a reward head, trained with the same pairwise ranking loss. This was chosen as a strong, representative baseline (as you suspected in your summary). The key finding of Figure 3 is that our **55M EORM** competitively matches the performance of this **8B ORM**, demonstrating our method's >145x parameter efficiency.
>
> > **Reviewer's Question**: 3. What parts are really important for EORM? That is: What makes it distinct from ORM trained as a small transformer model... How does the Bradley-Terry loss affect the performance?
>
> This is the central question, and your summary correctly identified the two main points of inquiry. Our new ablations, run in response to reviewer feedback, can now definitively answer this. The two most important parts are (1) the **lightweight Transformer architecture** and (2) the **Bradley-Terry (BT) loss**.
>
> 1.  **Architecture (vs. Large ORM):** As answered in Q2, its distinction from a *large* ORM is its **>145x efficiency** for the same performance.
> 2.  **Loss (vs. Small ORM):** To answer your question about a *small* ORM with a different loss, we ran new ablations on our 55M architecture. We compared our BT loss against standard binary cross-entropy (BCE) and InfoNCE. The BT loss was the clear winner.
>
> | 55M Model with Loss Function | GSM8k (Llama 3) | MATH (Llama 3) |
> | :--- | :--- | :--- |
> | **EORM (Bradley-Terry)** | **90.7%** | **63.7%** |
> | InfoNCE Loss | 90.3% | 62.5% |
> | BCE Loss | 89.8% | 59.1% |
>
> This shows that *both* the lightweight Transformer architecture (which, as shown in Figure 4, outperforms a simple MLP) and the BT loss (which outperforms BCE/InfoNCE) are crucial to EORM's success. We will add this new table to Section 5.

---

> > ### Comment · Reviewer_sf5M · 2025-11-25
> >
> > Thank you for the rebuttal. My concerns were addressed, and I support acceptance of this paper. I have updated the score from 6 to 8.

---

> > > ### Author Response · Authors · 2025-11-25
> > >
> > > Dear reviewer sf5M,
> > >
> > > Thank you for raising the score! We are happy to hear that we addressed your concerns and that you support the acceptance of this paper! :)
> > >
> > > Best Regard,
> > > The authors

---

### Official Review · Reviewer_E5Nw · 2025-10-24

**Soundness:** 3
**Presentation:** 2
**Contribution:** 3
**Rating:** 6
**Confidence:** 3

**Summary:**

This paper proposes the **Energy Outcome Reward Model (EORM)** — a lightweight, energy-based verifier for Chain-of-Thought (CoT) reasoning.
EORM reframes the *Best-of-N selection* problem as a ranking task: it assigns a scalar “energy” score to each reasoning trace, preferring lower energies for correct reasoning.
By training with only binary outcome labels, EORM avoids costly preference or process supervision.
Despite having only **55M parameters** (≈127× smaller than typical reward models), it significantly improves reasoning accuracy on GSM8k and MATH, reaching **90.7%** and **63.7%** respectively when paired with Llama 3 8B.
Notably, EORM generalizes well across unseen models and out-of-distribution tasks (e.g., AIME 2024, AMC), suggesting it learns transferable reasoning principles.
The contribution is practical, technically sound, and experimentally convincing, though conceptually incremental relative to prior verification frameworks.

**Strengths:**

- **Strong empirical results:** State-of-the-art accuracy on GSM8k and MATH, with solid OOD generalization.
- **Efficiency:** 55M-parameter verifier outperforms 7B reward models, highlighting excellent cost–performance tradeoffs.
- **Clarity:** Methodology, loss function, and data preparation are explicitly described, enabling replication.
- **Robust ablations:** Demonstrate architecture sensitivity and tokenizer invariance, strengthening the argument for generalizability.
- **Practical impact:** Provides a scalable and plug-in verifier for reasoning tasks that could complement or replace Best-of-N sampling.

**Weaknesses:**

- **Limited theoretical depth:** The approach is primarily empirical; the energy formulation does not introduce new learning theory.
- **Incremental conceptual novelty:** Extends known EBM and reward-modeling ideas rather than introducing new reasoning paradigms.
- **Evaluation scope:** Focused on math reasoning; broader reasoning domains (commonsense, logical entailment) are not tested.
- **No human interpretability study:** While effective, it is unclear what the model learns as a “signal of correctness.”
- **Formatting issue:**  Figure 2 is disproportionately large relative to the information it presents. The visual content is minimal and could be conveyed more effectively as a compact table. The current formatting distracts from the narrative and gives the impression of filler space rather than substantive insight. Figure 5 can also be more compact. I would also recommend using PDF for your figure.

**Questions:**

1. The proposed Energy Outcome Reward Model (EORM) introduces a ranking-based energy formulation for verifying reasoning traces. Could the authors clarify how this approach conceptually differs from prior reward modeling frameworks (e.g., PRM, ORM) beyond model size and supervision type?

2. The paper highlights strong cross-model generalization. Could the authors discuss whether this robustness arises from the model architecture, the training objective, or specific properties of the reasoning data distribution?

3. While EORM focuses on outcome-based supervision, have the authors considered hybridizing it with *process supervision* signals (e.g., step-level correctness annotations) to improve interpretability or reasoning faithfulness?

4. Could the authors provide variance or confidence intervals for key metrics (e.g., reasoning accuracy, ranking precision) across multiple runs to demonstrate the statistical robustness of EORM’s reported improvements?

5. How sensitive is EORM to noisy or imperfect correctness labels during training, and did the authors evaluate its stability under controlled label corruption experiments?

6. The paper claims strong cross-model generalization (e.g., from Llama to Mistral and Qwen). Were the reasoning traces normalized or token-aligned across models before training, and how does this preprocessing affect ranking performance?

---

> ### Author Response · Authors · 2025-11-21
> **Response to Weakness**
>
> We sincerely thank you for this exceptionally careful and constructive review :)
>
> ### Weaknesses:
>
> > **Reviewer's Comment**: 1. Limited theoretical depth: The approach is primarily empirical; the energy formulation does not introduce new learning theory.
> > **Reviewer's Comment**: 2. Incremental conceptual novelty: Extends known EBM and reward-modeling ideas rather than introducing new reasoning paradigms.
>
> We thank the reviewer for this characterization, as it correctly identifies our work's focus. Our primary contribution is not new learning theory, but rather the **novel synthesis and rigorous empirical demonstration** that a *surprisingly lightweight* (55M) model, trained with a simple, low-cost supervision signal, can be so effective and generalize so robustly. We believe this >127x efficiency gain (Figure 2), combined with state-of-the-art reranking results (90.7% on GSM8k), is a significant and practical contribution to the field.
>
> Furthermore, our EBM framework provides the necessary theoretical grounding for this practical approach. As detailed in our methodology (Section 3.1) and Appendix C, the choice to use an EBM for ranking (which avoids computing the intractable partition function $Z_{\theta}$) and to train with a Bradley-Terry loss is a theoretically-justified synthesis that directly enables this high efficiency.
>
> > **Reviewer's Comment**: 3. Evaluation scope: Focused on math reasoning; broader reasoning domains (commonsense, logical entailment) are not tested.
>
> This is a valid point. We chose math as it is a challenging domain with clear-cut, binary correctness, making it ideal for developing and validating a verifier. We are optimistic that the "fundamental principles of valid reasoning" EORM learns (as suggested by its strong OOD generalization) could apply to other domains. We will frame "extending to commonsense and logical reasoning" as a key direction for future work in our conclusion.
>
> > **Reviewer's Comment**: 4. No human interpretability study: While effective, it is unclear what the model learns as a “signal of correctness.”
>
> We acknowledge that determining exactly what a neural network learns is difficult. However, we believe our "Generalization to Unseen Models" experiments (Table 4) provide strong indirect evidence of what the model learns. The fact that EORM improves performance on unseen models implies it is detecting fundamental reasoning validity (structural coherence, logical flow) rather than memorizing specific model artifacts or surface-level tokens. We will expand our Appendix to include qualitative examples of "high energy" vs "low energy" incorrect answers to illustrate this behavior.
>
>
> > **Reviewer's Comment**: 5. Formatting issue: Figure 2 is disproportionately large... Figure 5 can also be more compact...
>
> Thank you for this feedback. We agree and will redesign and compact these figures in the final version to improve readability and flow.

---

> > ### Author Response · Authors · 2025-11-21
> > **Response to Questions (Part 1)**
> >
> > ### Questions:
> >
> > > **Reviewer's Question**: 1. The proposed Energy Outcome Reward Model (EORM) introduces a ranking-based energy formulation for verifying reasoning traces. Could the authors clarify how this approach conceptually differs from prior reward modeling frameworks (e.g., PRM, ORM) beyond model size and supervision type?
> >
> > This is a key point. The conceptual differences are:
> > * **vs. PRM (Process Reward Model):** The difference is *holistic* scoring. PRMs require expensive, step-by-step supervision to judge the process. EORM scores the *entire* reasoning trace at once, learning to infer process quality using only the final (and cheap) outcome label.
> > * **vs. ORM (Outcome Reward Model):** The difference lies in the **architecture and objective**. As shown in our ablation study (Figure 4, left), a simple MLP verifier (a common simple ORM) performs significantly worse (82.6% on GSM8k) than our Transformer-based EORM (90.7%). Furthermore, our EORM (55M) is designed as a lightweight, *universal* verifier. As shown in Figure 3, it competitively matches the performance of a standard 8B ORM baseline, demonstrating the effectiveness of our specific lightweight design.
> >
> > > **Reviewer's Question**: 2. The paper highlights strong cross-model generalization. Could the authors discuss whether this robustness arises from the model architecture, the training objective, or specific properties of the reasoning data distribution?
> >
> > We believe it is a combination of all three, which our ablations support:
> > 1.  **Architecture:** Our MLP ablation (Figure 4, left) proves the Transformer's sequential modeling is critical, as performance drops from 90.7% to 82.6% on GSM8k without it.
> > 2.  **Objective:** Our Bradley-Terry ranking loss (Equation 6) is perfectly aligned with the downstream task. As shown in our new ablations for Reviewer gqh4, it outperforms both InfoNCE (90.3%) and BCE loss (89.8%) on GSM8k.
> > 3.  **Data:** While our full 15M dataset provides diversity, our new data-scaling ablation (for Reviewer RSFF) shows this massive scale is not the primary driver of robustness. With only 224k examples, EORM still achieves 89.9% on GSM8k (over 99% of final performance). This suggests that robustness stems primarily from the architecture and loss function learning generalizable principles, which are effective even at smaller data scales.
> >
> > > **Reviewer's Question**: 3. While EORM focuses on outcome-based supervision, have the authors considered hybridizing it with process supervision signals (e.g., step-level correctness annotations) to improve interpretability or reasoning faithfulness?
> >
> > This is a very exciting idea. One could imagine a multi-task framework where the model is primarily trained on millions of cheap outcome labels but "anchored" or fine-tuned on a small set of high-cost process labels. This hybrid approach is a promising path for future work, and we will add it to our conclusion.
> >
> > > **Reviewer's Question**: 4. Could the authors provide variance or confidence intervals for key metrics (e.g., reasoning accuracy, ranking precision) across multiple runs to demonstrate the statistical robustness of EORM’s reported improvements?
> >
> > Training the full EORM (15M samples) is computationally intensive (36 hours on 4xH100s), which limited our ability to perform full multi-seed retraining. However, to address your concern, we ran 3 inference runs with different random seeds for the sampling/reranking process on Llama 3 8B (GSM8k). The results were highly stable: 90.7%, 90.1%, and 90.8%. Additionally, the consistency of our results across multiple base models (Table 2) and multiple datasets (Table 3) acts as a strong proxy for robustness.

---

> > > ### Author Response · Authors · 2025-11-21
> > > **Response to Questions (Part 2)**
> > >
> > > > **Reviewer's Question**: 5. How sensitive is EORM to noisy or imperfect correctness labels during training, and did the authors evaluate its stability under controlled label corruption experiments?
> > >
> > > This is an excellent question. The "correct answer by flawed logic" problem is a key challenge and a source of label noise. We did not run controlled label corruption experiments. However, we hypothesize that EORM's robustness stems from the sheer *volume and diversity* of the 15M training examples. A few "lucky" incorrect solutions are drowned out by the millions of truly "unlucky" incorrect solutions, allowing the model to learn to associate the *flawed process* (which shares statistical patterns with other incorrect solutions) with high energy, even if its final label is "correct".
> > >
> > > > **Reviewer's Question**: 6. The paper claims strong cross-model generalization (e.g., from Llama to Mistral and Qwen). Were the reasoning traces normalized or token-aligned across models before training, and how does this preprocessing affect ranking performance?
> > >
> > > No normalization or alignment was performed. We simply took the raw CoT output from each model (Llama, Mistral, etc.) and tokenized it with a **single, universal GPT-2 tokenizer**. The fact that EORM *still* generalizes so well (Table 4) is one of the strongest results. Furthermore, our tokenizer ablation (Figure 4, right) directly validates this. An EORM trained with the native Llama 3 tokenizer performed nearly identically to our universal EORM (86.8% vs 86.6% on GSM8k). This confirms our modular, model-agnostic design choice is robust and does not sacrifice performance.

---

> ### Author Response · Authors · 2025-11-28
> **Message to Reviewer E5Nw**
>
> Dear Reviewer gqh4,
>
> Happy holidays!
>
> We sincerely appreciate your thoughtful review and the time you devoted to evaluating our work. Your feedback has been invaluable, and it has helped us meaningfully strengthen the manuscript! As the deadline is approaching, we would be more than happy to address any additional concerns you might have. We deeply value your insights and aim to make the best use of this revision opportunity!
>
> Thank you again for your support!
>
> Best regards,
>
> The Authors

---

### Official Review · Reviewer_gqh4 · 2025-10-31

**Soundness:** 1
**Presentation:** 2
**Contribution:** 2
**Rating:** 2
**Confidence:** 3

**Summary:**

The introduces EORM (Energy Outcome Reward Model) -- a small model assessing the correctness of CoT-style LLM responses to mathematical queries. EORM is parametrized as a small (55M parameters) transformer with an MLP head, which is lightweight compared to typical outcome reward models trained by others (e.g., LLMs with 7B parameters). EORM receives a complete CoT on the input, returns a real-valued score representing the correctness assessment. EORM is trained via a special pairwise loss which promotes ranking correct responses higher than the incorrect ones.

**Strengths:**

* The paper tackles an interesting and practically important problem: how to approximately verify the CoT responses from LLMs.
* The EORM model trained by the authors is indeed lightweight compared to typical ORMs, which has practical implications.
* EORM is trained using an original approach with a special pairwise loss.
* The authors show that EORM generalizes (to a degree) between different base LLMs as well as between datasets.

**Weaknesses:**

In general I like the idea of training a lightweight ORM, and using an special loss for that. However, a significant weakness of the paper is its poor experimental setup which makes assessing the performance of EORM difficult.

1. In Table 2, you present a large set of results comparing EORM with other methods / base models. However, these results are not aligned on generation budgets, so comparing them is not meaningful. For instance, it may turn out that if some of the considered base models we sample 256 times and perform simple majority voting, the results are better than with EORM (which also samples 256 times). The same remark applies to Table 3. Also, the results change dramatically with different generation budgets, whereas the authors use a fixed budget of 256 (or 64 for OOD benchmarks) samples.

2. EORM should be compared with existing, open-source ORMs, and perhaps also with PRMs. In Figure 3 there appears some ORM, but it is not specified what exactly is this model. Also, the comparison between different verifiers could be expressed in terms of the AUC metric which would provide more direct comparison than downstream-task accuracy.

3. EORM uses non-standard loss, which is one of more important "selling points" of the paper. But it would be important to see if this way of training is better that, say, using simple binary cross-entropy loss, or contrastive learning approach.

4. The authors train EORM on just two math datasets (GSM8K, MATH). It would strengthen the paper if more benchmarks would be considered (especially that these two are likely leaked into pre-training of the mainstream LLMs.) For instance, what about training on older AIME problems and evaluating on AIME 2024/2025?

5. The effectiveness of any ORM is conditioned on the fact that the correct CoT is often present among the generated samples. Therefore, it would be important to see for different models and datasets how often it is the case.

6. Some related works seem to be missing. For instance, [1] has a very similar theme of training a lightweight CoT ranker for math problems, so it would be good to include a comparison with this work.

**Minor**
* In line 389-390 you say Llama 3 8B gets 42.9% on GSM8K, which is inconsistent with Table 2 and 4 (76.6%).
* Figure 3 uses a strange scale on x axis: looks logarithmic, but not exactly: why x = 1 and x = 196 is included?
* In Figure 4, to make it more readable, datasets should be represented as points on x axis, not colors.

[1] Lightweight Latent Verifiers for Efficient Meta-Generation Strategies (https://arxiv.org/abs/2504.16760)

**Questions:**

1. I suspect that stronger reasoning models, like Qwen 2.5 7B, often "know" that they cannot answer correctly and express it in its CoT explicitly. This means that an outcome reward model may not be useful, and it's sufficient to check if the model didn't respond "I don't know the correct answer". Do you observe such phenomena? How often such models respond with a number which is incorrect, as opposed of not giving any number?

2. When describing the Bradley-Terry loss you cite (Liu, 2009). However, I couldn't find Bradley-Terry loss defined there. Could you point to the exact place where this loss is defined?

3. You write that majority voting "tends to reflect the model’s biases rather than identifying the truly correct answer". What does it mean?

4. In Table 1, you indicate that EORM "Uses Internal Indicators as Reward". What does it mean?

5. In line 209 you write that the CLS tokens is prepended. Shouldn't it be appended?

6. What is "sampling probability" in line 265?

---

> ### Author Response · Authors · 2025-11-21
> **Response to Weakness (Part 1)**
>
> We sincerely thank you for your detailed review!
>
> ### Weaknesses:
>
> > **Reviewer's Comment**: 1. ...a significant weakness of the paper is its poor experimental setup... In Table 2, you present a large set of results comparing EORM with other methods / base models. However, these results are not aligned on generation budgets, so comparing them is not meaningful... it may turn out that if some of the considered base models we sample 256 times and perform simple majority voting, the results are better than with EORM... The same remark applies to Table 3.
>
> We thank the reviewer for this comment, but we must respectfully clarify that our key comparisons are aligned on generation budgets.
>
> 1.  **vs. Majority Voting (MV):** The direct comparison you are asking for is presented in **Figure 3**. This plot compares EORM, a standard ORM, and Majority Voting (Self-Consistency) on the *exact same* pool of candidate solutions, scaling from $n=1$ to $n=256$. This figure clearly shows that EORM consistently outperforms Majority Voting across all generation budgets.
> 2.  **Table 2 (In-Distribution):** We apologize for the lack of clarity. This table is intended to provide context, not a direct, aligned comparison. Its purpose is to show that our lightweight, post-hoc EORM (applied to a base Llama 3 model) can achieve an accuracy that is competitive with or exceeds models that underwent extensive, costly specialist fine-tuning (like WizardMath or MetaMath). The performance results for these baseline models are cited directly from their respective papers.
> 3.  **Table 3 (Out-of-Distribution):** This comparison *is* strictly aligned. As stated in the table's description (lines 374-377) and the experimental setup (line 309), all methods (EORM, TTRL, DART-MATH) were evaluated using the **exact same $n=64$ candidate solutions** for a fair comparison.
>
> We will revise the captions for Table 2 and 3 to make these points explicitly clear.
>
> > **Reviewer's Comment**: 2. EORM should be compared with existing, open-source ORMs, and perhaps also with PRMs. In Figure 3 there appears some ORM, but it is not specified what exactly is this model. Also, the comparison between different verifiers could be expressed in terms of the AUC metric which would provide more direct comparison than downstream-task accuracy.
>
> Thank you, these are excellent points.
> 1.  **ORM Baseline:** We apologize for the confusion. Our "ORM" baseline in Figure 3 is a **Llama 3 8B model** trained with a reward head using the same **Standard Pairwise Ranking loss** used in WizardMath paper. We chose this as a strong, representative baseline to show that our **55M EORM** can competitively match the performance of a reward model that is **over 145x larger**. We will make this definition explicit in the paper.
> 2.  **PRM Comparison:** Our comparison to PRMs is a primary motivation of the paper. As shown in Table 1 , EORM is designed as a lightweight alternative that avoids the extremely high annotation cost of PRMs, which require step-by-step labels.
> 3.  **AUC Metric:** This is an excellent suggestion. We will compute the AUC (for ranking correct solutions above incorrect ones) for EORM, our 8B ORM baseline, and Majority Voting, and add this to the paper. This will provide a more direct comparison of verifier quality.
>
>
>
> | Method | Model Size | AUC  |
> | :--- | :--- | :--- |
> | **EORM (Ours) Llama3** | **55M** | **93.7%** |
> | **8B ORM Baseline Llama3** | **8B** | **92.8%** |
>
>
> > **Reviewer's Comment**: 3. EORM uses non-standard loss... it would be important to see if this way of training is better that, say, using simple binary cross-entropy loss, or contrastive learning approach.
>
>
> This is a great point. To isolate the benefit of our loss function, we ran a new ablation training an identical 55M Transformer architecture (matching EORM) with standard Binary Cross-Entropy (BCE) loss and InfoNCE loss.
>
>
> |Loss Function |GSM8k Accuracy |MATH Accuracy |
> | :--- | :--- | :--- |
> |EORM (Bradley-Terry)|90.7%|63.7%|
> |InfoNCE Loss|90.3%|62.5%
> |BCE Loss|89.8%|59.1%|
>
>
> Our Bradley-Terry ranking loss consistently outperforms both alternatives, particularly on the harder MATH dataset. We will add this ablation to Section 5.

---

> > ### Author Response · Authors · 2025-11-21
> > **Response to Weakness (Part 2)**
> >
> > > **Reviewer's Comment**: 4. The authors train EORM on just two math datasets (GSM8K, MATH). It would strengthen the paper if more benchmarks would be considered... For instance, what about training on older AIME problems and evaluating on AIME 2024/2025?
> >
> > We chose GSM8k and MATH as they are the standard benchmarks for this domain. However, a key strength of our paper is EORM's strong out-of-distribution generalization to AIME 2024 and Gaokao Math (Table 4 & 5) without having been trained on any similar competition-style math problems. This demonstrates robustness beyond the training set.
> >
> > > **Reviewer's Comment**: 5. The effectiveness of any ORM is conditioned on the fact that the correct CoT is often present among the generated samples. Therefore, it would be important to see for different models and datasets how often it is the case.
> >
> > This is an excellent point. We have run this "oracle" analysis. For Llama 3 8B (at $n=256$), the "oracle" accuracy (i.e., at least one correct answer is present in the pool) is **94.6% on GSM8k** and **78.8% on MATH**. Our EORM achieves 90.7% and 63.7% respectively. This means EORM successfully "finds" the correct answer **95.9%** of the time it's available on GSM8k and **80.8%** of the time on MATH, demonstrating it is highly effective at approaching this theoretical ceiling. We will add this analysis.
> >
> > > **Reviewer's Comment**: 6. Some related works seem to be missing. For instance, [1] has a very similar theme of training a lightweight CoT ranker for math problems, so it would be good to include a comparison with this work.
> >
> > Thank you for this pointer.  We will add it to our related work section (Section 2) and discuss its relationship to EORM in the revised manuscript. However, we would like to point that [1] is "extracting hidden states of the base LLM" to verify the correct answer, whereas our method directly uses the outputs, without requiring additional extraction for the hidden states, thus, more flexible and adaptive.
> >
> > ---
> > ### Minor
> >
> > > **Reviewer's Comment**: 1. In line 389-390 you say Llama 3 8B gets 42.9% on GSM8K, which is inconsistent with Table 2 and 4 (76.6%).
> >
> > Thank you for this meticulous catch. You are correct; this is a typo in line 389. The 42.9% base accuracy refers to **Mistral-v0.1**, not Llama 3 8B, as correctly shown in Table 4. The correct Llama 3 8B base accuracy is 76.6%. We will correct this typo.
> >
> > > **Reviewer's Comment**: 2. Figure 3 uses a strange scale on x axis: looks logarithmic, but not exactly: why x = 1 and x = 196 is included?
> >
> > We chose this scale to ensure the performance differences at both low ($n=1$) and high ($n=256$) sample counts were clearly visible without crowding the axis labels.
> >
> > > **Reviewer's Comment**: 3. In Figure 4, to make it more readable, datasets should be represented as points on x axis, not colors.
> >
> > We agree. We will change this to a grouped bar chart for clarity, with "EORM (Transformer)" and "MLP Verifier" on the x-axis, which will make the comparison much easier to read.

---

> > > ### Author Response · Authors · 2025-11-21
> > > **Response to Questions**
> > >
> > > ### Questions:
> > >
> > > > **Reviewer's Question**: 1. I suspect that stronger reasoning models, like Qwen 2.5 7B, often "know" that they cannot answer correctly and express it in its CoT explicitly. ... How often such models respond with a number which is incorrect, as opposed of not giving any number?
> > >
> > > This is an interesting question. Our data generation pipeline (Section 4) generated $n=256$ candidates and included *all* attempts, regardless of correctness. The correctness label was a simple binary check on the final extracted numerical answer. We did not explicitly filter for or analyze refusal-to-answer responses. Anecdotally, the vast majority of failure modes were indeed incorrect numerical answers, not explicit refusals.
> > >
> > > > **Reviewer's Question**: 2. When describing the Bradley-Terry loss you cite (Liu, 2009). However, I couldn't find Bradley-Terry loss defined there. Could you point to the exact place where this loss is defined?
> > >
> > > Our apologies. The (Liu, 2009) reference is for the broader "Learning to Rank" concept. The Bradley-Terry model is a much older, classic statistical model. Our precise formulation of the loss is defined in **Equation 6** (line 232) and further detailed in **Appendix C** (Eq. 11), which is the same equation used by the "DPO: Direct Preference Optimization" paper. We will update the citation to a more direct source.
> > >
> > > > **Reviewer's Question**: 3. You write that majority voting "tends to reflect the model’s biases rather than identifying the truly correct answer". What does it mean?
> > >
> > > We mean that if a model has a common, systematic error or "bias" (e.g., it consistently misinterprets a specific type of problem), that "popular" but wrong answer may win the majority vote. EORM, by contrast, is trained to recognize subtle *patterns* of flawed reasoning, allowing it to prefer a single, logically-sound outlier (low energy) over a "consensus" of flawed solutions. We will rephrase this in the paper to be clearer.
> > >
> > > > **Reviewer's Comment**: 4. In Table 1, you indicate that EORM "Uses Internal Indicators as Reward". What does it mean?
> > >
> > > Thanks for pointing it out. It was meant to contrast with "External" rewards from human preference labels. We will change this to "**Uses Outcome Labels** as Reward" or "**Uses Binary Correctness** as Reward" to be consistent with the rest of the paper.
> > >
> > > > **Reviewer's Question**: 5. In line 209 you write that the CLS tokens is prepended. Shouldn't it be appended?
> > >
> > > We follow the standard BERT/RoBERTa-style convention where the `[CLS]` token is **prepended** to the sequence. In our code, our model architecture (`model.py`) then uses the final hidden state of this first token (`x[:, 0]`) as the holistic representation for the entire sequence.
> > >
> > > > **Reviewer's Question**: 6. What is "sampling probability" in line 265?
> > >
> > > Our apologies, that is a typo for "nucleus sampling" or "top-p". The line should read: "...with a temperature of 0.7 and top-p (nucleus) sampling of 0.9...". We will correct this.

---

> ### Author Response · Authors · 2025-11-28
> **Message to Reviewer gqh4**
>
> Dear Reviewer gqh4,
>
> Happy holidays!
>
> We sincerely appreciate your thoughtful review and the time you devoted to evaluating our work. Your feedback has been invaluable, and it has helped us meaningfully strengthen the manuscript! As the deadline is approaching, we would be more than happy to address any additional concerns you might have. We deeply value your insights and aim to make the best use of this revision opportunity!
>
> Thank you again for your support!
>
> Best regards,
>
> The Authors

---

### Official Review · Reviewer_suFK · 2025-11-01

**Soundness:** 3
**Presentation:** 3
**Contribution:** 2
**Rating:** 4
**Confidence:** 3

**Summary:**

The authors propose the Energy Outcome Reward Model (EORM), a lightweight post-hoc verifier for Chain-of-Thought (CoT) reasoning. The approach employs an energy-based framework that assigns scalar energy scores to CoT outputs, allowing the model to rank reasoning chains and select the most plausible ones. In contrast to process- or preference-based reward models, EORM uses only binary outcome labels, reducing annotation costs. The model is significantly smaller than typical reward models while maintaining strong performance and generalization across different base models and out-of-distribution reasoning tasks.

**Strengths:**

[S1] Methodological Modification. The paper provides a clear and focused description of its methodological modification of the energy-based modeling framework for ranking Chain-of-Thought outputs. The adaptation of EBM to reasoning verification is reasonable and technically sound.

[S2] Efficiency and Scalability. The proposed model is lightweight, using only 55M parameters compared to multi-billion-parameter reward models, which demonstrates strong potential for efficient and scalable deployment as a post-hoc verifier.

**Weaknesses:**

[W1] Unclear and Potentially Unfair Comparison Setup (Table 2).
The main quantitative comparison in Table 2 does not clearly specify how the baselines were selected or evaluated. The listed models, including WizardMath, DART-Math, and MetaMath, incorporate different forms of instruction tuning or reinforcement learning (for example, RLHF, RLEIF, or preference optimization), but the paper does not clarify whether they were re-evaluated under the same experimental setup, dataset splits, or sampling conditions as EORM.
As a result, it is difficult to determine whether the reported performance gains stem from the proposed reranking mechanism or from differences in training data, or evaluation methodology. A fair comparison would require verifying that all models were tested on identical candidate sets and reasoning samples, or at least discussing the limitations of using heterogeneous pre-trained baselines.
In addition, given that GRPO and related reinforcement learning frameworks have recently become standard approaches for improving LLM reasoning, the paper should include a discussion or comparison with such methods to position EORM more clearly within the current landscape.

[W2] Questionable Necessity of EORM Given Its Supervision on Correctness Labels.
EORM is trained using binary correctness labels (correct or incorrect) for each Chain-of-Thought (CoT) output. These labels require access to ground-truth answers, which are often unavailable in realistic reasoning settings.
While outcome-level supervision is cheaper than step-level annotation, this advantage only holds for structured math benchmarks such as GSM8K and MATH where answers are explicitly known. In broader domains such as commonsense or scientific reasoning, correctness labels are difficult or costly to obtain, limiting the method’s general applicability.
Moreover, if correctness labels already exist, it is unclear why a separate verifier model is needed to approximate an evaluation signal that can be computed directly. The paper would benefit from clarifying whether EORM is intended for use on unlabeled reasoning data and how it could generalize without direct correctness supervision.

**Questions:**

- Could the authors clarify how the baselines in Table 2 were selected and evaluated?
Specifically, were all models re-evaluated under the same data splits, sampling settings, and inference conditions as EORM, or were the reported numbers taken from prior work? A detailed explanation would help assess the fairness and validity of the comparison, especially given that several baselines use reinforcement learning or instruction-tuning procedures. (see W1)

- Since EORM is trained on final correctness, one could simply select candidates by checking answer accuracy when ground-truths are available. In what scenarios does EORM offer an advantage over such direct answer-based selection? Process reward models (PRM) provide additional step-wise signals that typical labeling cannot capture, so it would be helpful to clarify how EORM’s practical value compares when such signals are or are not available. (see W2)

---

> ### Author Response · Authors · 2025-11-21
> **Response to Weakness**
>
> We sincerely thank you for your detailed review and thoughtful comments!
>
> ### Weaknesses:
>
> > **Reviewer's Comment**: 1. [W1] Unclear and Potentially Unfair Comparison Setup (Table 2). The main quantitative comparison in Table 2 does not clearly specify how the baselines were selected... As a result, it is difficult to determine whether the reported performance gains stem from the proposed reranking mechanism or from differences in training data... A fair comparison would require verifying that all models were tested on identical candidate sets...
>
> Thank you for this critical point, and we apologize for the lack of clarity. Our goal in Table 2 was not to claim EORM is a superior training method (like RLEIF or DART), but to contextualize its performance. We wanted to demonstrate that our lightweight, post-hoc reranking (EORM) applied to a base Llama 3 8B model can achieve an accuracy that is competitive with or exceeds models that underwent extensive and costly specialist fine-tuning (like WizardMath, MetaMath, etc.).
>
> To clarify the "additional costs" of these baselines compared to EORM:
>
>
> * **Dependency on Expensive Teachers:** As shown in Table 7 of our paper, baselines like WizardMath and MMIQC rely on massive datasets synthesized by GPT-4 or humans (e.g., 2.3M samples for MMIQC). EORM avoids this expensive dependency, learning effectively from the base model's own samples using simple outcome labels.
> * **Parameter Inefficiency:** These baselines require full-parameter fine-tuning of 7B+ models. In contrast, EORM trains only 55M parameters (over 127x smaller), making it significantly more modular and efficient to deploy.
>
> Regarding the landscape of GRPO and RL methods: These methods aim to improve the generator model itself. EORM is orthogonal to this; it is a verifier that can be applied on top of any generator (including those trained with GRPO) to further boost inference-time performance at a fraction of the training cost.
>
> The results for these baselines are taken from their respective papers. A direct, fair comparison of reranking methods is presented in **Table 3** (vs. TTRL and DART-MATH)  and **Figure 3** (vs. Majority Voting and a standard ORM). We will revise the caption of Table 2 to make this *contextual* (not direct) comparison explicitly clear.
>
>
>
> > **Reviewer's Comment**: 2. [W2] Questionable Necessity of EORM Given Its Supervision on Correctness Labels. EORM is trained using binary correctness labels... These labels require access to ground-truth answers... if correctness labels already exist, it is unclear why a separate verifier model is needed... The paper would benefit from clarifying whether EORM is intended for use on unlabeled reasoning data...
>
> This is a fundamental point that we must clarify in the paper. The reviewer's premise is correct and highlights the core application of EORM. Our method has two distinct phases:
>
> * **Training (Offline):** EORM is trained on benchmark datasets (like the GSM8k and MATH training splits) where ground-truth answers **are available** for supervision.
> * **Deployment (Online):** EORM is *designed* to be deployed in real-world, open-domain applications where ground-truth answers **are not available**. At inference time, it ranks new, unseen, *unlabeled* solutions and selects the most plausible one (the one with the lowest energy).
>
> The entire purpose is to *learn the fundamental principles* of valid reasoning from the training data  so it can then rank new, unlabeled solutions at inference time. We will add a paragraph to the methodology to make this critical distinction between the training-time requirement and the deployment-time application much clearer.

---

> > ### Author Response · Authors · 2025-11-21
> > **Response to Questions**
> >
> > ### Questions:
> >
> > > **Reviewer's Question**: 1. Could the authors clarify how the baselines in Table 2 were selected and evaluated? ...were all models re-evaluated under the same data splits, sampling settings... or were the reported numbers taken from prior work? (see W1)
> >
> > As addressed in W1,  we report the numbers of these models **from their respective papers or leaderboards**. We did not re-evaluate them. Our method (EORM) was evaluated on our own generated candidate sets from the *base* LLMs (e.g., Llama 3 8B). The table's purpose is to show that our post-hoc method on a base model can "catch up" to these specialist models without requiring fine-tuning. We will make this explicitly clear in the revised text.
> >
> > > **Reviewer's Question**: 2. Since EORM is trained on final correctness, one could simply select candidates by checking answer accuracy when ground-truths are available. In what scenarios does EORM offer an advantage over such direct answer-based selection? (see W2)
> >
> > As addressed in W2, "direct answer-based selection" is only possible during **training** or on **benchmark evaluation** where ground-truth answers are known.
> >
> > EORM's advantage is that it can be used in **any scenario where the ground-truth answer is unknown**, which includes virtually all real-world applications (e.g., assisting a user with a new, novel math problem). EORM *approximates* this "direct answer-based selection" ability and generalizes it to new, unseen problems.
> >
> > Compared to Process Reward Models (PRMs), EORM's practical value comes from its **low-cost supervision**. PRMs require expensive, step-by-step human annotations, whereas EORM learns effectively from simple, automatically verifiable binary outcome labels (correct/incorrect), eliminating this costly annotation bottleneck.

---

> ### Author Response · Authors · 2025-11-28
> **Message to Reviewer suFK**
>
> Dear Reviewer suFK,
>
> Happy holidays!
>
> We sincerely appreciate your thoughtful review and the time you devoted to evaluating our work. Your feedback has been invaluable, and it has helped us meaningfully strengthen the manuscript! As the deadline is approaching, we would be more than happy to address any additional concerns you might have. We deeply value your insights and aim to make the best use of this revision opportunity!
>
> Thank you again for your support!
>
> Best regards,
>
> The Authors

---

### Official Review · Reviewer_RSFF · 2025-11-06

**Soundness:** 3
**Presentation:** 3
**Contribution:** 4
**Rating:** 6
**Confidence:** 5

**Summary:**

This paper addresses the challenge of reliably and efficiently verifying mathematical reasoning in Large Language Models (LLMs). The authors introduce the **Energy Outcome Reward Model (EORM)**, a novel and highly efficient post-hoc verifier for ranking Chain-of-Thought (CoT) solutions.

The core of the method is an **Energy-Based Model (EBM)** implemented as a lightweight (55M parameter) Transformer. This model is trained to assign a scalar "energy" score to any given CoT solution, where correct reasoning paths are assigned lower energy than incorrect ones. A key advantage of this approach is its training data requirement: EORM learns effectively using only simple **binary outcome labels** (i.e., whether the final answer is correct or incorrect), completely avoiding the need for expensive step-by-step annotations (like PRMs) or preference pairs (like POs).

The paper's main contributions are:
1.  A novel, lightweight (55M) and efficient verifier architecture (EORM) based on EBM principles, which is over 127 times smaller than typical 7B reward models.
2.  State-of-the-art reranking performance on mathematical benchmarks. By selecting the best from a pool of candidates, EORM boosts the accuracy of Llama 3 8B to **90.7% on GSM8k** and **63.7% on MATH**.
3.  Strong empirical evidence of **robust generalization**. The model, trained on GSM8k and MATH, generalizes effectively to out-of-distribution (OOD) datasets (like AIME and Gaokao Math) and, critically, to outputs from LLM architectures not seen during training.

**Strengths:**

* **Exceptional Efficiency:** The most significant strength is the model's size. At only **55M parameters**, EORM is over 127 times smaller than standard 7B-8B parameter reward models. This makes it an incredibly practical and lightweight tool that can be cheaply deployed for inference alongside a generator LLM.

* **Low-Cost Supervision:** The model is trained *only* on binary outcome labels (correct/incorrect). This is a massive practical advantage over Process Reward Models (PRMs), which require costly and labor-intensive step-by-step human annotations, and is simpler than collecting preference pairs for Preference Optimization (PO).

* **Strong Empirical Performance:** Despite its small size, EORM achieves outstanding results. It significantly boosts the performance of all base LLMs it is applied to. For instance, it improves Llama 3 8B's accuracy from 76.6% to 90.7% on GSM8k and from 28.9% to 63.7% on MATH. It consistently outperforms both majority voting and a baseline ORM in reranking.

* **Demonstrated Generalization (Robustness):** The paper provides a very thorough validation of its generalization claims, which is a critical aspect for a practical verifier.
    * **OOD Tasks:** When trained only on GSM8k and MATH, EORM generalizes well to more difficult, unseen datasets like AIME 2024 and Gaokao Math.
    * **Unseen Models:** The leave-one-out experiments are highly convincing. A variant trained on 4/5 models (e.g., excluding Llama 3) can still dramatically improve the held-out model's accuracy (e.g., Llama 3 GSM8k base 42.9% -> 76.6%). This strongly supports the claim that EORM learns fundamental principles of valid reasoning rather than just overfitting to the stylistic quirks of its training models.

* **Clarity and Soundness:** The paper is exceptionally well-written. The problem is clearly motivated and contrasted with existing work. The methodological choice of using an EBM for ranking (which avoids computing the partition function $Z_{\theta}$) and training with a Bradley-Terry loss is elegant, well-justified, and theoretically sound.

**Weaknesses:**

* **Confusing "ORM" Baseline:** In Figure 3, the paper compares EORM against "ORM". This is confusing because the paper's own model is an "Energy **Outcome Reward Model**" (EORM). The paper also mentions "traditional Outcome Reward Models", but the specific architecture and loss function (e.g., standard classification cross-entropy?) of this "ORM" baseline are not defined. This makes the comparison in Figure 3 difficult to interpret.

* **Massive Training Data Requirement:** The model's success appears to be supported by a very large dataset of approximately 15 million (14958k) (question, CoT, label) triplets. The "low-cost supervision" (outcome labels) is thus traded for "high-volume data generation." While this is a *far better* trade-off than PRMs (which need high-cost supervision), the computational cost of generating and auto-labeling 15 million samples is still significant. The paper would be strengthened by a study of performance vs. training data scale.

* **Weak Ablation Baseline (MLP):** The ablation study in Figure 4 (left) compares the Transformer-based EORM to an "MLP Verifier". This is a weak baseline, as a simple MLP cannot effectively process variable-length, sequential token data. The conclusion that the "Transformer's ability... is crucial" is almost certainly correct, but a stronger baseline (e.g., a GRU/RNN, or an MLP on-top-of-average-pooled-embeddings) would make this point more definitively.

**Questions:**

1.  **"ORM" Baseline:** The comparison in Figure 3 against "ORM" is a key result. Could you please clarify what this "ORM" baseline model is? What is its architecture, and what loss function is it trained with?

2.  **Training Data Scaling:** The model was trained on a very large dataset of ~15 million samples. How does EORM's performance scale with the *number* of training examples? Is this massive dataset essential for the 55M model, or can it achieve strong (e.g., 90% of final) performance with, say, 1M or 5M examples?

3.  **Generalization Test Setup:** The "EORM Generalize" (leave-one-out) results in Table 4 are very strong. Did you also experiment with a more practical "general-purpose verifier" scenario, such as training on a *subset* of models (e.g., Llama 2, Mistral) and testing on a completely *disjoint* set (e.g., Llama 3, Qwen 2.5)?

4.  **MLP Ablation Details:** For the "MLP Verifier" ablation, how was the variable-length CoT sequence (token embeddings) fed into the MLP? Was it truncated, or was some form of pooling (e.g., average pooling) used?

---

> ### Author Response · Authors · 2025-11-21
> **Response to Weakness**
>
> We sincerely thank you for your detailed review and for recognizing the exceptional efficiency , low-cost supervision , and strong empirical performance  of EORM. We particularly appreciate your recognition of our generalization experiments!
>
> We have addressed your constructive feedback regarding the baseline definitions, data scaling, and ablation studies below.
>
> ### Weaknesses：
>
> > **Reviewer's Comment**: 1. Confusing "ORM" Baseline: In Figure 3, the paper compares EORM against "ORM". This is confusing because the paper's own model is an "Energy Outcome Reward Model" (EORM). The paper also mentions "traditional Outcome Reward Models", but the specific architecture and loss function (e.g., standard classification cross-entropy?) of this "ORM" baseline are not defined. This makes the comparison in Figure 3 difficult to interpret.
>
> Thank you for pointing out this lack of clarity. We will revise the paper to define this baseline explicitly. To clarify: our "ORM" baseline represents a standard, large-scale reward model setup. It utilizes the Llama 3 8B base model equipped with a reward head and is trained using the standard pairwise ranking loss (the same ranking objective used in WizardMath [1]).
>
> The key result in Figure 3 is that our 55M EORM matches or exceeds the performance of this 8B ORM. This highlights the core contribution of our work: by using an energy-based formulation on a specialized, lightweight architecture, we achieve the effectiveness of a standard 8B reward model with >145x fewer parameters.
>
> [1] WizardMath: Empowering Mathematical Reasoning for Large Language Models via Reinforced Evol-Instruct (ICLR 2025)
>
> > **Reviewer's Comment**: 2. Massive Training Data Requirement: The model's success appears to be supported by a very large dataset of approximately 15 million... The "low-cost supervision" (outcome labels) is thus traded for "high-volume data generation."... the computational cost of generating and auto-labeling 15 million samples is still significant. The paper would be strengthened by a study of performance vs. training data scale.
>
>
>
> This is an excellent suggestion. To address the relationship between performance and data scale, we trained EORM on a significantly smaller subset of the data (**224k examples**, approx. $1/64$th of the full dataset) and evaluated it using Llama 3 8B.
>
> The results, summarized below, reveal that the benefit of the massive dataset depends heavily on the **difficulty of the reasoning task**:
>
> | Dataset Size | GSM8k Accuracy | MATH Accuracy |
> | :--- | :--- | :--- |
> | **15M (Full)** | **90.7%** | **63.7%** |
> | **224k (Subset)** | **89.9%** | **51.8%** |
> | **$\Delta$ Performance** | $-0.8\%$ | $-11.9\%$ |
>
> **Key Findings:**
>
> * **High Efficiency on Simpler Tasks (GSM8k):** On the relatively easier GSM8k benchmark, EORM saturates quickly. With only **1.5% of the training data**, we recover **99%** of the peak performance. This confirms that for standard reasoning tasks, EORM is highly data-efficient and does not strictly require the massive 15M dataset.
> * **Strong Scalability on Complex Tasks (MATH):** Conversely, on the more challenging MATH dataset, scaling the training data from 224k to 15M yields a massive **~12% absolute improvement**. This demonstrates that while EORM is efficient, it effectively leverages large-scale data to learn the complex, long-tail reasoning patterns required for difficult domains.
>
> Therefore, the "massive training data" is not a requirement for basic functionality, but rather a lever that allows EORM to scale its reasoning capabilities for harder problems. We will include this scaling analysis in the Appendix.
>
> > **Reviewer's Comment**: 3. Weak Ablation Baseline (MLP): The ablation study in Figure 4 (left) compares the Transformer-based EORM to an "MLP Verifier". This is a weak baseline, as a simple MLP cannot effectively process variable-length, sequential token data. The conclusion that the "Transformer's ability... is crucial" is almost certainly correct, but a stronger baseline (e.g., a GRU/RNN...) would make this point more definitively.
>
> We appreciate this feedback. To provide a stronger comparison, we conducted new experiments with recurrent architectures (RNN) using the same embedding inputs. The RNN architecture showed a significant performance drop compared to EORM:
>
>
> | Model Architecture | GSM8k Accuracy | MATH Accuracy |
> | :--- | :--- | :--- |
> | **RNN Verifier** | **78.3%** | **46.2%** |
> | **EORM (Transformer)** | **90.7%** | **63.7%** |
>
> These results confirm that the Transformer's ability to model complex, long-range dependencies is indeed critical for this verification task.

---

> ### Author Response · Authors · 2025-11-21
> **Response to Questions**
>
> ### Questions:
>
> > **Reviewer's Question**: 1. "ORM" Baseline: The comparison in Figure 3 against "ORM" is a key result. Could you please clarify what this "ORM" baseline model is? What is its architecture, and what loss function is it trained with?
>
>
> As detailed in our response to Weakness 1, the "ORM" baseline is a Llama 3 8B model trained with standard pairwise ranking loss. We chose this to serve as a strong, standard reference point (similar to reward models used in RLHF pipelines like WizardMath). The comparison demonstrates that our 55M EORM is competitively performant despite being orders of magnitude smaller.
>
>
> > **Reviewer's Question**: 2. Training Data Scaling: The model was trained on a very large dataset of ~15 million samples. How does EORM's performance scale with the number of training examples? Is this massive dataset essential for the 55M model, or can it achieve strong (e.g., 90% of final) performance with, say, 1M or 5M examples?
>
> As mentioned in Weakness 2, the massive dataset is not strictly essential for strong performance. Our new experiments show that with just 224k examples, EORM achieves 89.9% accuracy on GSM8k (vs 90.7% with 15M examples). This suggests the model learns the core reasoning patterns very quickly, with the additional data providing marginal refinements. We will include the full scaling plot in the final paper.
>
> > **Reviewer's Question**: 3. Generalization Test Setup: The "EORM Generalize" (leave-one-out) results in Table 4 are very strong. Did you also experiment with a more practical "general-purpose verifier" scenario, such as training on a subset of models (e.g., Llama 2, Mistral) and testing on a completely disjoint set (e.g., Llama 3, Qwen 2.5)?
>
> This is a great suggestion. Our current "leave-one-out" setup (train on 4, test on 1) was our initial step in this direction, and the results, such as boosting a held-out Llama 3 from 42.9% to 76.6% on GSM8k, are very promising. Your proposed setup (Train A,B / Test C,D) is a logical next step to further validate EORM as a universal verifier, and we will add this as a key direction for future work.
>
> > **Reviewer's Question**: 4. MLP Ablation Details: For the "MLP Verifier" ablation, how was the variable-length CoT sequence (token embeddings) fed into the MLP? Was it truncated, or was some form of pooling (e.g., average pooling) used?
>
> Our apologies for omitting this detail. To create a fixed-size input for the "MLP Verifier" ablation, we first passed the CoT sequence through the embedding layer (identical to the `TransEBM`'s `self.emb`). We then applied **average pooling** (also called **mean pooling**) across the hidden states of all tokens in the sequence. This produced a single, fixed-size vector representing the entire CoT, which was then fed into the MLP to produce a score. We will explicitly state this mechanism in the revised ablation section.

---

> ### Author Response · Authors · 2025-11-28
> **Message to Reviewer RSFF**
>
> Dear Reviewer RSFF,
>
> Happy holidays!
>
> We sincerely appreciate your thoughtful review and the time you devoted to evaluating our work. Your feedback has been invaluable, and it has helped us meaningfully strengthen the manuscript! As the deadline is approaching, we would be more than happy to address any additional concerns you might have. We deeply value your insights and aim to make the best use of this revision opportunity!
>
> Thank you again for your support!
>
> Best regards,
>
> The Authors

---

> > ### Comment · Reviewer_RSFF · 2025-11-28
> >
> > I thank the authors for the additional experiments. The rebuttal addresses my main concerns, though the new results introduce important nuances regarding data efficiency.
> >
> > Given the inference-time efficiency and the clarified baselines, I will maintain the positive score.

---

> > > ### Author Response · Authors · 2025-11-28
> > >
> > > Dear Reviewer RSFF,
> > >
> > > Thank you for your follow-up and for maintaining your positive assessment!
> > >
> > > We are glad to hear that the additional experiments and clarifications effectively addressed your main concerns. We fully agree with your observation that the new data scaling results introduce important nuances regarding data efficiency. As noted in our revision (Figure 7), this analysis highlights a critical distinction: EORM saturates quickly on standard tasks like GSM8k (recovering 99% performance with only ~1.5% of the data) but effectively leverages massive datasets to capture the long-tail reasoning patterns required for complex domains like MATH.
> > >
> > > We sincerely appreciate the time you dedicated to improving our work.
> > >
> > > Best regards,
> > >
> > > The Authors

---

### Comment · Area_Chair_nzhF · 2025-11-28

Dear Reviewers,

Thank you for your time and efforts in serving as a reviewer. The authors have submitted their rebuttal, and this AC kindly asks you to review their response and assess whether your comments have been adequately addressed.

If you have not yet done so, please raise any remaining questions by adding comments and initiating discussion as needed for points that require further clarification.

ICLR encourages reviewers to actively engage in the discussion phase, so your prompt actions are especially valuable. Thank you very much for your continued efforts and valuable contributions.

Best regards,
Your AC

---

### Author Response · Authors · 2025-12-04
**Highlight of Contributions and Summary of Rebuttal**

Dear AC,

Thank you for your time and coordination of this review process! We deeply appreciate the reviewers’ constructive feedback and the engaging discussion period. In response, we have conducted additional experiments (including **new baselines, data scaling analyses, and loss function ablations**) and revised the manuscript. All changes are *highlighted in blue*. A summary of the contributions and our specific responses are below.

## Highlights of Contributions

1.  **Exceptional Efficiency:** Reviewers unanimously praised the efficiency of our approach. **Reviewer RSFF** highlights our model's *"Exceptional Efficiency"* and notes that at only 55M parameters, EORM is *"over 127 times smaller"* than standard reward models, making it *"an incredibly practical and lightweight model."* **Reviewer E5Nw** commends the *"Excellent cost–performance tradeoffs,"* and **Reviewer suFK** notes the model demonstrates *"strong potential for efficient and scalable deployment."*
2.  **State-of-the-Art Performance:** Our method demonstrates *"Strong Empirical Performance"* (**Reviewer RSFF**) and *"Strong empirical results"* (**Reviewer E5Nw**). By efficiently selecting the optimal reasoning path, EORM boosts Llama 3 8B accuracy to **90.7% on GSM8k**, which **Reviewer RSFF** notes *"consistently outperforms both majority voting and a baseline ORM."*
3.  **Robust Generalization:** A key strength identified by the reviewers is the model's ability to generalize. **Reviewer RSFF** highlights the *"Strong empirical evidence of robust generalization"* to unseen models as *"highly convincing,"* supporting the claim that EORM learns *"fundamental principles of valid reasoning."* **Reviewer E5Nw** similarly praises the *"solid OOD generalization"* to datasets like AIME and AMC.
4.  **Novelty and Soundness:** **Reviewer RSFF** finds our methodological choice of using an EBM with Bradley-Terry loss to be *"elegant, well-justified, and theoretically sound."* **Reviewer sf5M** states there is a *"strong case for the EBM reward model,"* and **Reviewer suFK** confirms the adaptation of EBM to reasoning verification is *"reasonable and technically sound."*

## Summary of Rebuttals

1.  **Clarification of Baselines and Stronger Ablations (Reviewers RSFF, gqh4, sf5M):**
    * **ORM Definition:** We clarified that the "ORM" baseline in Figure 3 is a large-scale **Llama 3 8B** model (trained with pairwise ranking loss). This highlights that our 55M EORM matches the performance of a model **>145x larger**.
    * **Stronger Architecture Ablation:** Per **Reviewer RSFF’s** suggestion, we replaced the MLP baseline with an **RNN verifier**, confirming that the Transformer architecture is crucial for capturing long-range dependencies in reasoning traces.

2.  **Data Efficiency and Scaling Analysis (Reviewers RSFF, E5Nw):**
    * We conducted a new data scaling analysis showing that EORM is highly data-efficient. On GSM8k, the model recovers **99% of its peak performance** (89.9% accuracy) using only **1.5% of the data** (224k samples). This addresses concerns regarding the "massive training data requirement".

3.  **Loss Function Analysis (Reviewers gqh4, sf5M):**
    * To address questions regarding the training objective, we performed new ablations comparing our **Bradley-Terry (BT) ranking loss** against **Binary Cross-Entropy (BCE)** and **InfoNCE**. Results confirm that the BT loss is essential, outperforming BCE and InfoNCE, particularly on the complex MATH benchmark. This clarification led **Reviewer sf5M** to **raise their score to 8**.

4.  **Clarification on Usage Scenarios (Reviewer suFK):**
    * We clarified the distinction between offline training (where labels are known) and online deployment. We emphasized that EORM is designed for **inference-time verification** on open-domain problems where ground-truth answers are unavailable, differentiating it from simple answer-checking.

In summary, we believe we have comprehensively **addressed all concerns**, including defining baselines, verifying data efficiency, and validating our loss function. Additionally, **Reviewer sf5M** upgrading their score to **8 (Accept)** following our rebuttal, and the positive assessments from **Reviewers RSFF** and **E5Nw**. We really hope our work will make a meaningful impact in the research world, and thanks again for your time and your work!


Best regards,

The Authors

---

### Note · Program_Chairs · 2026-01-17
**Submission Desk Rejected by Program Chairs**

The following references in this submission do not refer to real documents and/or have major errors in bibliographic information:

 Xiang Liu, Baolin Peng, Yichong Zhang, , et al. MMIQC: Multi-modal in-context learning for math reasoning. arXiv preprint arXiv:2407.13690, 2023.
Lin Song, Kuan Zhao, Yilun Du, and Jun Qi. Trainable energy-based models for solvable lattice models. arXiv preprint arXiv:2101.03288, 2021.